# SpikingLM: Towards Fully Spiking Language Model

Yu Liang [1]   Zijian Zhou [1]   Wenjie Wei [1]   Shuai Wang [1]   Honglin Cao [1]   Ammar Belatreche [2]
Yu Yang [1]   Malu Zhang [1 3]   Yang Yang [1]   Haizhou Li [3 4]

## Abstract

Leveraging event-driven computation mechanism, Spiking Neural Networks (SNNs) have emerged as a representative paradigm for energy-efficient edge intelligence. However, extending SNNs to modern deep language models still faces two fundamental challenges. First, dead neurons in deep SNNs lead to degraded gradients, limiting the training effectiveness of spiking language models. Second, removing Softmax for energy efficiency weakens token-wise competition, reducing the model's ability to select salient tokens. To address these challenges, we propose Spiking Language Model (SpikingLM) to bridge the efficiency of SNNs and the capability of modern language models through two key innovations. First, we propose Distribution-aware Scaling method, which rescales linear outputs into an activation-friendly range to alleviate dead neurons and stabilize gradient propagation. Notably, its scaling parameters can be fused into the preceding linear layers, incurring no additional inference overhead. Second, we introduce Spike2Max to restore winner-takes-all mechanism via base-2 exponentiation and max-subtraction. Compared with Softmax, Spike2Max reduces energy consumption by over 95% using hardware-efficient bit-shift operations. Experiments show that SpikingLM reduces energy consumption by 57.9% and achieves state-of-the-art performance on GLUE, laying a promising foundation for energy-efficient language modeling. Code is available at https://github.com/hamings1/SpikingLM.git.

[1]University of Electronic Science and Technology of China [2]Northumbria University [3]Shenzhen Loop Area Institute [4]The Chinese University of Hong Kong (Shenzhen). Correspondence to: Shuai Wang <wangshuai718@std.uestc.edu.cn>.

*Proceedings of the 43$^{rd}$ International Conference on Machine Learning*, Seoul, South Korea. PMLR 306, 2026. Copyright 2026 by the author(s).

## 1. Introduction

Large language models (LLMs) have fundamentally transformed natural language processing, achieving remarkable performance across diverse tasks from text generation to complex reasoning (Brown et al., 2020; Touvron et al., 2023; Achiam et al., 2023). However, this success comes at substantial computational and environmental costs. Modern LLMs require billions of multiply-accumulate (MAC) operations per token, resulting in significant energy consumption that raises concerns about sustainability and limits deployment on resource-constrained devices (Patterson et al., 2021; Strubell et al., 2019). As LLMs become increasingly integral to everyday applications, the need for energy-efficient inference architectures has never been more pressing.

Spiking Neural Networks (SNNs) (Maass, 1997; Gerstner & Kistler, 2002) have gained significant attention due to their brain-inspired dynamics (Izhikevich, 2003; Masquelier et al., 2008). Spiking neurons emit discrete spikes only when activated and remain silent otherwise. Compared with conventional Artificial Neural Networks (ANNs) dominated by costly MAC operations, the spike-driven mechanism in SNNs enables lightweight accumulate (AC) operations, thereby reducing computational energy cost. This advantage becomes particularly pronounced on neuromorphic platforms such as Tianjic (Pei et al., 2019) and Loihi (Davies et al., 2018; Orchard et al., 2021). Motivated by these benefits, SNNs have been extensively studied in computer vision tasks, including image classification (Rathi et al., 2020; Fang et al., 2021a; Deng et al., 2022; Zheng et al., 2021), object detection (Luo et al., 2024), and semantic segmentation (Kim et al., 2022; Yao et al., 2025; Wang et al., 2026).

Despite remarkable advances in SNNs, extending them to natural language processing remains a formidable challenge. Recently, SpikeBERT (Lv et al., 2025) and SpikeLM (Xing et al., 2024) have made promising strides toward spike-based language models. Nevertheless, these approaches still exhibit a noticeable performance gap compared to their ANN counterparts (Devlin et al., 2019; Brown et al., 2020), and fail to fully preserve spike-driven mechanism. In this work, we identify two fundamental challenges in spiking language models. First, through empirical investigation, we find that as network depth increases, the probability of sustained

neuronal inactivation rises progressively. These dead neurons induce severe gradient degradation (Bellec et al., 2018) during backpropagation, fundamentally compromising the model's representational capacity (Rathi et al., 2020; Fang et al., 2021b). Second, the absence of softmax (Vaswani et al., 2017) in Spike Self-attention (SSA) (Zhou et al., 2022; 2026; Yao et al., 2023) causes attention scores to exhibit diffuse distributions, leading to a marked decline in token selectivity and consequently impairing the model's ability to attend to semantically critical tokens (Katharopoulos et al., 2020).

In this paper, we propose a fully spiking language model (SpikingLM) as shown in Fig. 1, a framework that bridges the efficiency of SNNs with the capabilities of language models through targeted solutions for each identified challenge. To address gradient degradation, we introduce Distribution-aware Scaling method, a training technique that employs learnable scaling factors to prevent gradient vanishing. To restore attention selectivity, we propose a hardware-efficient Spike2Max attention that creates explicit competition through base-2 exponentiation and max-subtraction. Building on these two methods, SpikingLM achieves competitive accuracy and improved energy efficiency on GLUE benchmark (Wang et al., 2018), paving the way for sustainable language modeling on edge devices. The main contributions are as follows:

- **Problem analysis.** We identify the fundamental challenges in scaling SNNs to language tasks by establishing a fully spiking baseline and conducting performance gap analysis. Two core issues are pinpointed: gradient degradation from dead neurons and reduced token selectivity in attention mechanisms.

- **Distribution-aware Scaling**. We introduce learnable distribution-aware factors to rescale linear outputs into an activation-friendly range, thereby mitigating dead neurons and stabilizing gradient propagation. These factors can be fully fused into preceding linear layers at inference, incurring no additional overhead.

- **Spike2Max attention.** We propose an efficient attention mechanism that restores winner-takes-all dynamics via base-2 exponentiation and max-subtraction. Leveraging integer spike counts, Spike2Max replaces costly exponentials with bit shifts, enabling fine-grained token selectivity with lower hardware cost.

- **Empirical Validation.** Based on these innovations, we develop SpikingLM, a fully spiking language model that preserves strong language modeling capability. Extensive experiments show that SpikingLM achieves state-of-the-art performance on the GLUE benchmark and reduces energy consumption by 57.9% compared with conventional SNNs.

## 2. Related Work

**Training Mechanisms for SNNs.** Training SNNs poses significant theoretical and practical challenges. The discrete, non-differentiable nature of spike generation prevents application of backpropagation algorithms (Tavanaei et al., 2019). Surrogate gradient methods have emerged as the dominant training paradigm, approximating the Heaviside step function with continuous surrogates during backpropagation (Zenke & Ganguli, 2018; Shrestha & Orchard, 2018; Liang et al., 2025a). Deep SNNs face additional training difficulties, particularly gradient vanishing across network layers (Bellec et al., 2018; Wei et al., 2023; Zhang et al., 2021). The sparse firing patterns characteristic of efficient SNNs cause these issues, leading to dead neurons that provide no learning signals (Rathi et al., 2020; Fang et al., 2021b; Liang et al., 2025b). Due to these training complexities, SNN research has primarily concentrated on image classification (Rathi et al., 2020; Fang et al., 2021a; Deng et al., 2022; Li et al., 2021; Zheng et al., 2021; Wei et al., 2024), extending to complex tasks including object detection (Luo et al., 2024), semantic segmentation (Kim et al., 2022), and neuromorphic auditory perception (Tang et al., 2024; Wang et al., 2025b). While SNN research has flourished in computer vision (Rathi et al., 2020; Luo et al., 2024), progress in NLP remains limited. Recent models like SpikeBERT (Lv et al., 2025) and SpikeLM (Xing et al., 2024; Zhang et al., 2025) face a critical dilemma: they either lag behind ANNs in performance or retain expensive floating-point operations, failing to achieve fully spike-driven efficiency.

**Spiking Transformers and Attention.** Adapting Transformers to SNNs involves significant architectural challenges (Yao et al., 2025; Wang et al., 2025a; Xiao et al., 2025; Wei et al., 2026). Current spiking attention mechanisms often replace the softmax function with spike accumulation. However, this change prevents the model from creating competitive attention distributions, making it difficult for architectures like Spikformer to prioritize important tokens (Zhou et al., 2022; Zhang et al., 2026). This issue is similar to problems in linear attention, where the absence of exponential normalization leads to a "flat" attention matrix that fails to distinguish key features from background noise (Katharopoulos et al., 2020; Choromanski et al., 2020; Wang et al., 2026). This limitation is generally less pronounced in vision tasks, where the number of tokens is relatively limited and salient information is often spatially concentrated. In contrast, this limitation is more critical in natural language processing (Devlin et al., 2019; Brown et al., 2020), where the lack of focused token selection prevents SNNs from effectively capturing complex token interactions. Therefore, there is a critical need to develop specialized SNN mechanisms that can enhance token selectivity while maintaining the energy-efficient nature of spike-driven computation.

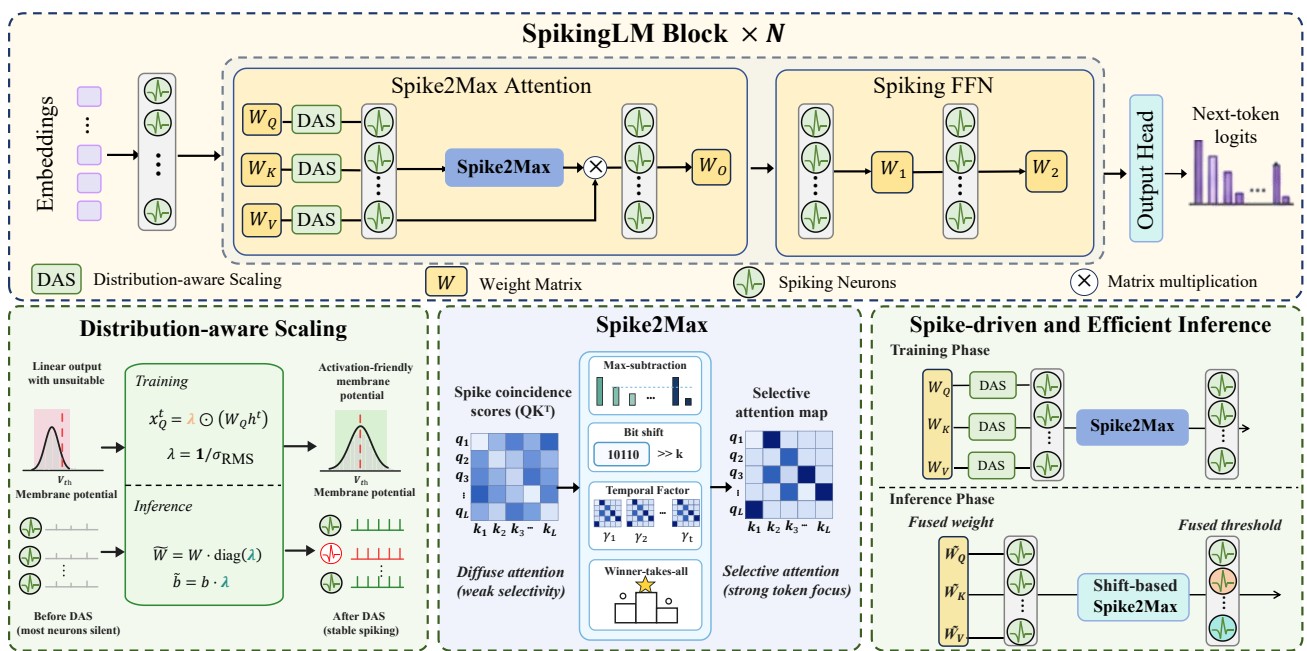

*Figure 1.* Overall pipeline of SpikingLM. Distribution-aware Scaling rescales activation distributions during training and is absorbed into the weights at inference, enabling improved optimization without additional inference overhead. Spike2Max enhances token selectivity through shift-based operations while avoiding complex floating-point computations.

## 3. Method

### 3.1. Problem Formulation

**Fully Spiking Language Model Baseline.** To maximize the energy efficiency of spiking language model, we construct a baseline that eliminates floating-point operations from the standard language model architecture. Specifically, we remove: (1) the softmax normalization in attention, replacing it with direct spike counting; (2) the GELU activation in feed-forward networks, substituting it with spiking neurons; and (3) all floating-point intermediate representations, constraining activations to discrete spikes. This yields a fully spiking language model where computation relies predominantly on addition operations.

*Spike encoding in language model.* SNNs encode information through discrete spikes rather than continuous activations. The Leaky Integrate-and-Fire (LIF) neuron (Gerstner et al., 2014) is the most widely used model, governed by:

$$m^t = \beta v^{t-1} + x^t, \tag{1}$$

$$s^t = \Theta(m^t - V_{th}), \tag{2}$$

$$v^t = m^t - V_{th} \cdot s^t, \tag{3}$$

where $v^t$ denotes the membrane potential, $x^t$ is the input, $\beta$ is the decay factor, and $V_{th}$ is the firing threshold. We adopt soft reset, where the membrane potential subtracts the threshold after firing. The spike output $s^t \in \{0, 1\}$ is determined by the Heaviside step function $\Theta(\cdot)$, which is

non-differentiable. During backpropagation, we employ the sigmoid function as surrogate gradient.

*Spiking Self-Attention in language model.* The standard Transformer (Vaswani et al., 2017) computes attention via softmax$(QK^\top / \sqrt{d_k})V$. To achieve fully spike-driven attention, we employ LIF neurons to transform the query, key, and value into binary spikes $S_Q, S_K, S_V \in \{0, 1\}^{T \times N \times d}$, where $T$ denotes the number of time steps. The spiking self-attention is then formulated as:

$$\text{Attention}(S_Q, S_K, S_V) = S_Q S_K{}^\top S_V, \tag{4}$$

where the softmax normalization is removed and all matrix multiplications are converted to additions, due to the inefficiency of exponential and division operations in softmax on neuromorphic hardware.

### 3.2. Performance Gap Analysis

Despite achieving significant computational savings, this baseline exhibits a notable performance gap compared to conventional language models. Our analysis identifies two core challenges:

#### Challenge 1: Gradient Degradation.

The non-differentiable nature of spike, combined with characteristically low firing rates in deep networks, leads to severe gradient vanishing during backpropagation. As the network depth increases, the magnitude of the input to spiking neurons diminishes, resulting in *dead spiking neurons*—

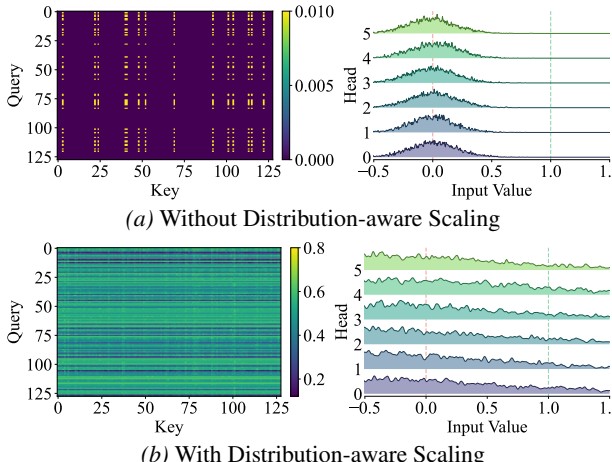

*(a)* Without Distribution-aware Scaling

*(b)* With Distribution-aware Scaling

*Figure 2.* Visualization of attention maps and membrane potential distributions. **(a)** Without scaling, the model exhibits extensive dead neuron regions (dark areas). **(b)** With scaling, neurons maintain effective membrane potential.

neurons that never fire and thus provide no gradient signals. As illustrated in Fig. 2a, models exhibit substantial inactive regions, severely degrading representational capacity.

*Challenge 2: Reduced Token Selectivity.*

Conventional spiking attention computes weights by counting spike coincidences between binary queries and keys, lacking the competitive mechanism inherent in softmax. We define token selectivity as the ability of an attention mechanism to concentrate weights on critical tokens while suppressing irrelevant ones. In traditional Transformers, softmax exponentially amplifies logit differences through $\text{softmax}(z)_i = \exp(z_i)/\sum_j \exp(z_j)$, concentrating attention on relevant positions. In contrast, SSA produces diffuse attention patterns with reduced token selectivity.

These challenges hinder the scalability and expressiveness of spiking architectures, creating a significant performance gap compared to their ANN counterparts. To bridge this gap, the following subsections present our solutions: Distribution-aware Scaling (§3.3) addresses the training instability, while Spike2Max (§3.4) resolves the reduced token selectivity issues.

### 3.3. Distribution-aware Scaling

To address **Challenge 1** (gradient degradation from dead neurons), we require scaling before spiking neurons. However, conventional approaches conflict with efficiency goals.

**Limitations of Conventional Scaling** Modern Large Language Models predominantly employ Layer Normalization (LN) (Ba et al., 2016) or Root Mean Square Normalization (RMSNorm) (Zhang & Sennrich, 2019), as adopted

by LLaMA (Touvron et al., 2023; Grattafiori et al., 2024) and DeepSeek (Guo et al., 2025). Standard RMSNorm computes:

$$\text{RMSNorm}(x) = \frac{x}{\sqrt{\frac{1}{d}\sum_{i=1}^{d} x_i^2 + \epsilon}} \cdot \gamma, \quad (5)$$

where $x \in \mathbb{R}^d$ is the input vector and $\gamma \in \mathbb{R}^d$ is the learnable scale parameter. However, this normalization requires instance-wise computation of the root-mean-square statistic. Such dynamic computation introduces division and square root operations that depend on the input $x$, incurring substantial MAC operations. This fundamentally conflicts with the *multiply-free paradigm* of SNNs. To overcome this, we replace dynamic normalization with learnable scaling factors that require no runtime statistics.

**Training forward propagation.** To resolve **Challenge 1** while preserving SNN efficiency, we introduce learnable scaling factors that rescale linear layer outputs to an optimal range, preventing gradient vanishing and maintaining firing rates. Specifically, we apply a learnable vector $\lambda \in \mathbb{R}^d$ to the linear projection outputs before the LIF neurons. Taking the query branch as an example, the input $x^t$ to the LIF neuron (as defined in Eq. 1) is modified as:

$$x_Q^t = \lambda \odot (W_Q \mathbf{h}^t), \quad (6)$$

where $\odot$ denotes element-wise multiplication and $\mathbf{h}^t$ is the hidden state at time step $t$. This scaling mechanism ensures that the membrane potential $m^t$ remains within a range conducive to generating stable spikes $S_Q$. To ensure stable learning, we initialize $\lambda$ based on the root-mean-square (RMS) statistics of the initial hidden states:

$$\lambda_{init} = \frac{1}{\sigma_{\text{RMS}}} \cdot \mathbf{1}, \quad \text{where} \quad \sigma_{\text{RMS}} = \sqrt{\frac{1}{d}\sum_{i=1}^{d} \mathbb{E}[h_i^2]}. \ (7)$$

The term $\sigma_{\text{RMS}}$ is estimated from the first training batch to normalize the input variance. This initialization ensures that the LIF neurons operate within an optimal firing range from the onset of training, facilitating faster convergence. In contrast to LayerNorm or RMSNorm, which necessitate per-sample statistical computation, our method facilitates learning process with *zero runtime overhead*. As illustrated in Fig. 2b, the integration of Distribution-aware Scaling maintains appropriate firing rates which are essential for guaranteeing effective gradient backpropagation.

**Inference Optimization.** A key advantage of our approach is that the learnable scaling factor $\lambda$ does not introduce any computational overhead during deployment. At inference, the scaling operation can be mathematically fused into the preceding linear transformation. Specifically, for

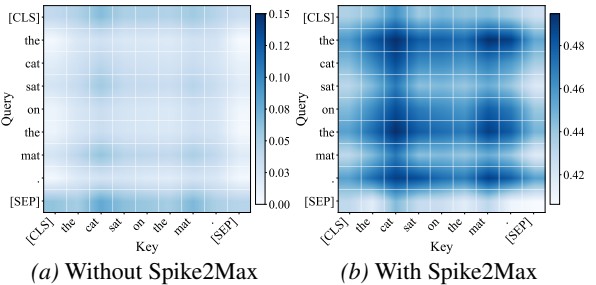

*(a)* Without Spike2Max     *(b)* With Spike2Max

*Figure 3.* Visualization of token selectivity in the attention mechanism. **(a)** Without Spike2Max, the attention responses exhibit reduced token selectivity, limiting the model's ability to focus on discriminative tokens. **(b)** With Spike2Max, salient token responses are preserved, resulting in enhanced token selectivity.

a linear layer defined by $y = Wx + b$, we construct the reparameterized weights $\tilde{W}$ and bias $\tilde{b}$ as follows:

$$\tilde{W} = W \cdot \text{diag}(\lambda), \quad \tilde{b} = b \cdot \lambda. \tag{8}$$

This fusion completely removes the scaling operation from inference, resulting in **zero additional MAC operations** for normalization while maintaining the benefits of representations learned during training. Consequently, our method resolves the dead neuron problem during training without compromising SNN efficiency at inference.

### 3.4. Spike2Max: Restoring Winner-Takes-All Attention

**The Missing Competition in Spiking Attention** Standard SSA computes attention weights from spike coincidence counts between binary-valued queries and keys. However, this formulation lacks the competitive **winner-takes-all** mechanism inherent to softmax attention. In conventional language models, the softmax function exponentially amplifies differences between logits, causing attention to concentrate sharply on the most relevant positions. In contrast, as shown in Fig. 3a, vanilla SSA produces relatively diffuse attention patterns, limiting the model's ability to selectively focus on salient tokens.

**Spike2Max** To address **Challenge 2**, we propose Spike2Max hardware-efficient mechanism that restores winner-takes-all dynamics in spiking attention. Spike2Max creates competition between attention positions through base-2 exponentiation and max-subtraction:

$$\text{Spike2Max}(A) = \gamma_t \cdot 2^{(A - \max(A) - 1)}, \tag{9}$$

where $A = QK^\top$ is the spike coincidence matrix, $\max(A)$ denotes row-wise maximum values, and $\gamma_t \in \mathbb{R}^T$ are learnable per-timestep scaling parameters.

**Winner-Takes-All Mechanism** The max-subtraction operation creates explicit competition: positions with the highest scores receive weight $2^{-1} = 0.5$, while other positions

*Table 1.* Energy consumption of softmax variants (45nm, 32-bit, row length $N$).

| Variant | Denominator | Operations | Energy |
|---------|-------------|------------|--------|
| Standard | $\sum_j e^{z_j}$ | $N$ exp, $N$ div | $38.9N$ pJ |
| Base-2 | $\sum_j 2^{z_j}$ | $N$ shift, $N$ div | $19.0N$ pJ |
| Spike2Max | $2^{\max_j z_j}$ | $N$ shift, $N$ cmp | $1.9N$ pJ |

obtain exponentially suppressed weights $2^{-1-\Delta}$, where $\Delta > 0$ represents the gap from the winner. This restores the token selectivity missing in naive spiking attention. The constant offset $-1$ controls the magnitude of unnormalized outputs. Since we remove the softmax denominator $\sum_j \exp(z_j)$ to avoid division operations, this offset provides a conservative baseline scale. The learnable parameter $\gamma_t$ subsequently adjusts the actual scale to task requirements. This strategy proves more stable than directly learning absolute magnitudes in vanilla SSA.

**Hardware-Efficient Implementation** Beyond its functional advantages, Spike2Max admits highly efficient neuromorphic implementation. In SNN formulation, queries and keys $Q, K \in \{0,1\}^{T \times d_k}$ are binary-valued, yielding integer-valued attention logits $QK^\top$ that correspond to spike coincidence counts. This integer structure enables exact computation of base-2 exponentials via left bit-shifting:

$$2^n = 1 \ll n, \quad n \in \mathbb{Z}_{\geq 0}, \tag{10}$$

entirely eliminating floating-point exponentiation. Furthermore, the max-subtraction normalization avoids the costly division operation required by softmax's denominator $\sum_j \exp(z_j)$. Tab. 1 quantifies these savings: assuming 45nm technology with 32-bit operations (Horowitz, 2014), Spike2Max reduces energy consumption by over 95% compared to standard softmax.

**Learnable Temporal Parameters** The per-timestep parameters $\gamma_t$ serve dual purposes: (1) **sharpness control**, regulating the peakedness of attention distributions; (2) **temporal adaptation**, learning the timestep-specific patterns inherent to SNNs. During inference, $\gamma_t$ can be folded into the firing thresholds of subsequent neurons via $V'_{th} = V_{th}/\gamma_t$, effectively achieving reparameterization that preserves the energy efficiency.

**Theorem 3.1** (Spike2Max Attention Approximation)**.** *Let $z \in \mathbb{R}^n$ denote attention logits and $V \in \mathbb{R}^{n \times d_v}$ denote value vectors. Define the softmax attention weights $p_i = softmax(z)_i$ and the Spike2Max weights $\hat{w}_i = \gamma_t \cdot 2^{z_i - \max(z) - 1}$, where $\gamma_t > 0$ is a learnable temporal scale*

*Table 2.* Comparison on the GLUE benchmark. SpikingLM is evaluated against ANN-based and other SNN baselines. Energy consumption is measured based on FP32 operations. SpikingLM is implemented with SpikingJelly (Fang et al., 2023) and is fully spiking, removing floating-point spikes, softmax, and GELU. **Bold** indicates the best result.

| Model | Time | MNLI$_{m/mm}$ | QQP$_{F1}$ | QNLI | SST-2 | CoLA | STS-B | MRPC$_{F1}$ | RTE | Avg. | Energy$_{mJ}$ |
|---|---|---|---|---|---|---|---|---|---|---|---|
| BERT$_{base}$ | – | – | 90.5 | 90.7 | 92.3 | 60.0 | 89.4 | 89.8 | 69.3 | 83.2 | 51.41 |
| BERT$_{3L}$ | – | 77.1/77.1 | 85.2 | 85.8 | 88.1 | 31.7 | 85.7 | 86.4 | 66.4 | 75.9 | 12.9 |
| Q2BERT | – | 47.2/47.3 | 67.0 | 61.3 | 80.6 | 0.0 | 4.7 | 81.2 | 52.7 | 49.1 | – |
| ELMo | – | 68.6/– | 86.2 | 71.1 | 91.5 | 44.1 | 70.4 | 76.6 | 53.4 | 70.2 | – |
| Spikingformer | 4 | 71.9/72.5 | 84.7 | 76.0 | 87.2 | 24.4 | 54.5 | 79.7 | 55.6 | 66.8 | 6.76 |
| SpikeBERT | 4 | 71.6/71.9 | 68.1 | 66.4 | 85.2 | 20.4 | 26.5 | 82.9 | 57.6 | 59.8 | 14.30 |
| SpikeLM | 4 | 77.1/77.2 | 83.9 | 85.3 | 87.0 | 38.8 | 84.9 | 85.7 | 69.0 | 76.5 | 13.74 |
| Ours | 4 | 75.5/75.4 | 84.4 | 84.6 | 90.1 | 50.4 | 82.4 | 87.9 | 61.7 | **77.1** | **5.79** |

*factor. The attention output approximation error satisfies:*

$$\left\| \sum_{i=1}^{n} p_i v_i - \sum_{i=1}^{n} \hat{w}_i v_i \right\|_2 \le (1 - \ln 2)\, \sigma_p(z) \, \|V\|_F \tag{11}$$
$$+ \mathcal{O}\big((1 - \ln 2)^2\big),$$

*where* $\sigma_p(z) = \sqrt{\sum_i p_i (z_i - \mu_p)^2}$ *is the softmax-weighted standard deviation of logits with* $\mu_p = \sum_i p_i z_i$. *For SNN attention with binary spike representations* $Q, K \in \{0,1\}^{T \times d_k}$, *this simplifies to:*

$$\mathcal{E}^* \le (1 - \ln 2)\sqrt{d_k} \approx 0.307\sqrt{d_k}. \tag{12}$$

The complete proof of Theorem.3.1 is provided in Appendix A. The above analysis shows that Spike2Max provides a principled approximation to softmax attention while remaining well aligned with the spike-driven computation.

## 4. Experiments

### 4.1. Experimental Setup

**Implementation.** SpikingLM is a fully spiking Transformer model derived from the BERT architecture. In particular, Softmax in self-attention, GELU (Hendrycks & Gimpel, 2016) in feed-forward networks, and dependencies on floating-point spike representations are eliminated. We train the model using sigmoid-like surrogate gradients with a time-step of $T = 4$. The LIF neurons are configured with a decay factor $\beta = 0.5$ and a threshold $V_{th} = 1.0$.

**Pre-training.** The model is pre-trained on large-scale unlabeled English corpora. The pre-training dataset consists of a mixture of publicly available text sources, including STORIES (Trinh & Le, 2018), BookCorpus (Zhu et al., 2015), CC-News (Hamborg et al., 2017), OpenWebText (Gokaslan et al., 2019), and English Wikipedia. Standard text preprocessing pipelines consistent with BERT pre-training (Devlin

et al., 2019) are adopted. Pre-training is performed using the masked language modeling objective, where a subset of input tokens is dynamically masked and the model is trained to predict the corresponding original tokens. During pre-training, optimization is performed using the AdamW optimizer with an initial learning rate of $2 \times 10^{-4}$. The model is trained for a total of 800,000 update steps with 5,000 warm-up steps. More details and dataset descriptions are shown in Appendix B.

**Fine-tuning.** To evaluate downstream performance, SpikingLM is fine-tuned on the GLUE benchmark (Wang et al., 2018), which includes a variety of natural language understanding tasks spanning sentence classification, sentence-pair classification, and regression. During fine-tuning, all model parameters are updated end-to-end using task-specific supervision. For classification tasks, a task-specific classifier is applied to the output representation, while regression tasks are optimized using a mean squared error objective.

**Baselines and Evaluation Metrics.** SpikingLM is compared against ANNs (Devlin et al., 2019; Liu et al., 2019) and SNNs (Lv et al., 2025) baselines with comparable architectural configurations and parameter scales. All baseline models are fine-tuned and evaluated using the same data splits and evaluation protocols to ensure fair comparison. Model performance is measured using the official GLUE evaluation metrics (Wang et al., 2018), including accuracy, F1 score, Matthews correlation coefficient, and Pearson or Spearman correlation, depending on the task.

### 4.2. Main Results

As illustrated in Tab. 2, we evaluate SpikingLM against a comprehensive set of ANN and SNN baselines. For the ANN benchmarks, we include BERT (Devlin et al., 2019), ELMo (Peters et al., 2018), and Q2BERT (Zhang et al., 2020). Furthermore, we compare our model with SNN ar-

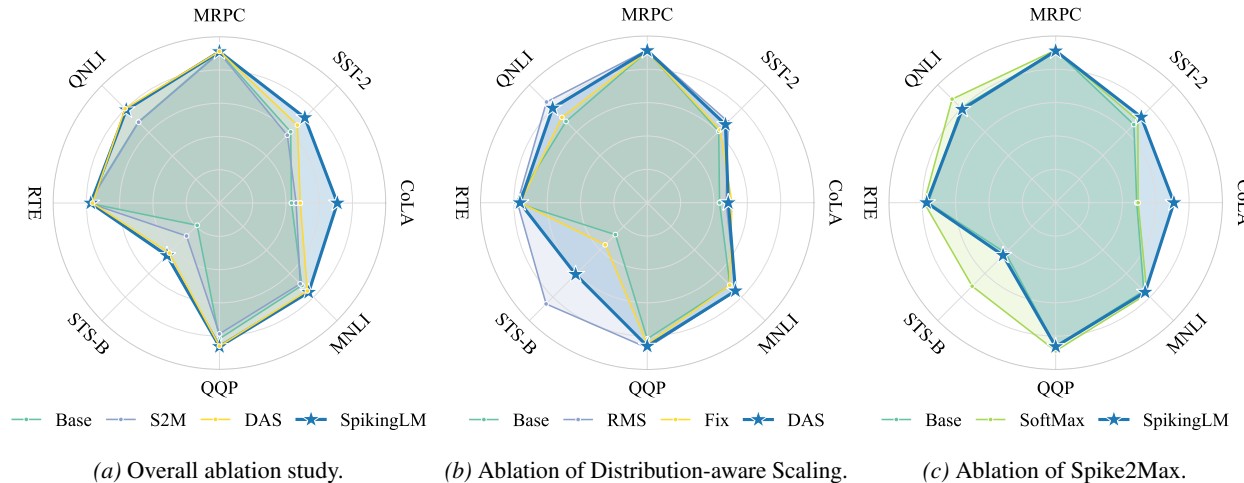

*(a)* Overall ablation study.  *(b)* Ablation of Distribution-aware Scaling.  *(c)* Ablation of Spike2Max.

*Figure 4.* Ablation results on GLUE evaluated with radar plots. The figure reports the effects of the proposed modules under different ablation settings, including overall ablation, Distribution-aware Scaling ablation, and Spike2Max ablation.

chitectures, specifically Spikingformer (Zhou et al., 2026), SpikeBERT (Lv et al., 2025), and SpikeLM (Xing et al., 2024). Compared with BERT$_{base}$, Ours achieves an average score of 77.1 while reducing inference energy from 51.41 mJ to 5.79 mJ, corresponding to an 88.7% reduction in energy consumption. Despite this substantial energy saving, Ours maintains competitive performance on several core tasks, including QNLI (84.6), SST-2 (90.1), and MRPC (87.9). Compared with Spikingformer, SpikingLM yields an improvement of 10.3 in the average score, increasing it from 66.8 to 77.1. Under the same inference time of 4 steps, Ours also reduces energy consumption from 6.76 mJ to 5.79 mJ, corresponding to a 14.3% energy reduction. Compared to SpikeBERT, SpikingLM achieves a 29% improvement in average GLUE score while reducing energy consumption by 8.51 mJ. We compare SpikingLM with SpikeLM, which relies on floating-point spikes and an energy-intensive softmax. Ours improves the average score from 76.5 to 77.1 while reducing energy consumption from 13.74 mJ to 5.79 mJ, resulting in a 57.9% reduction in energy. These results indicate a more efficient utilization of spiking activity without compromising overall task performance.

### 4.3. Ablation Studies

For the ablation study, we conduct ablation studies on Distribution-aware Scaling and Spike2Max, followed by a more fine-grained ablation of their internal components. The results on GLUE are summarized in the radar plot shown in Fig. 4a. SpikingLM incorporating Distribution-aware Scaling and Spike2Max consistently achieves competitive performance across all datasets. Notably, on CoLA, SpikingLM improves over the base model by 8.27. We further observe that Distribution-aware Scaling significantly enhances the learning capacity of SpikingLM, leading to consistent

improvements over the base model on GLUE. In contrast, Spike2Max yields more pronounced gains on tasks, which can be attributed to its improved token selectivity. Fig. 5a illustrates the evaluation loss during the pre-training stage. The results show that Distribution-aware Scaling enables faster model convergence, upon which SpikingLM further achieves improved convergence behavior.

**Ablation of Distribution-aware Scaling** For the ablation of Distribution-aware Scaling, we evaluate several variants, including the base model with RMSNorm and the base model with a fixed scaling factor set to 2, to analyze the effect. Fig. 4b indicates that using Distribution-aware Scaling is highly beneficial for language model training. However, RMSNorm introduces significant computational overhead due to the root mean square calculation. This additional overhead limits its suitability for deployment on neuromorphic devices. Fig. 5b shows that while fixed scaling aids convergence, its lack of adaptivity results in inferior final performance. In contrast, Distribution-aware Scaling introduces adaptivity during training, resulting in improved convergence accuracy, while avoiding the additional computational overhead associated with RMS-based methods through reparameterization.

**Ablation of Spike2Max** For the ablation study of Spike2Max, we adopt the model trained with Distribution-aware Scaling as the base and compare it with a variant augmented with softmax. As shown in Fig. 4c, although the softmax variant achieves a higher score, the introduced exponential operations and floating-point divisions incur non-negligible overhead. A more detailed energy analysis is provided in Section 4.7. In contrast, SpikingLM achieves a favorable trade-off between accuracy and energy efficiency. The evaluation loss during training is illustrated in Fig. 5c,

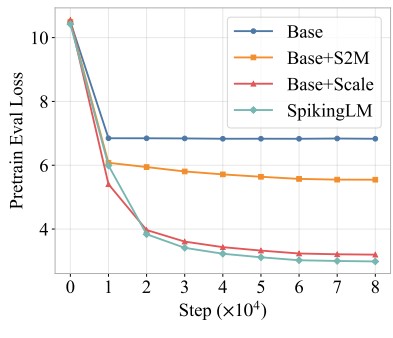 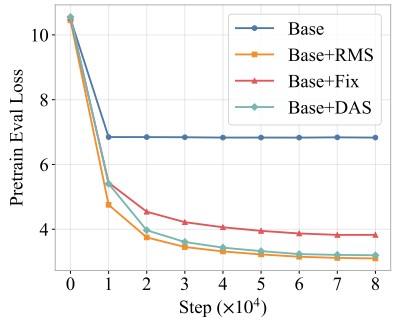 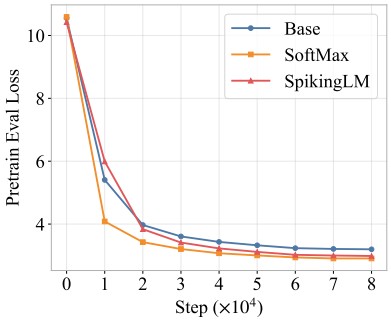

*(a)* Overall ablation study.    *(b)* Ablation of Distribution-aware Scaling.    *(c)* Ablation of Spike2Max.

*Figure 5.* Pre-training convergence under different ablation settings. The evaluation loss curves illustrate the impact of different modules on training dynamics, including overall ablation, Distribution-aware Scaling (DAS) ablation, and Spike2Max(S2M) ablation.

where SpikingLM attains convergence accuracy comparable to softmax and further improves over the variant using Distribution-aware Scaling alone.

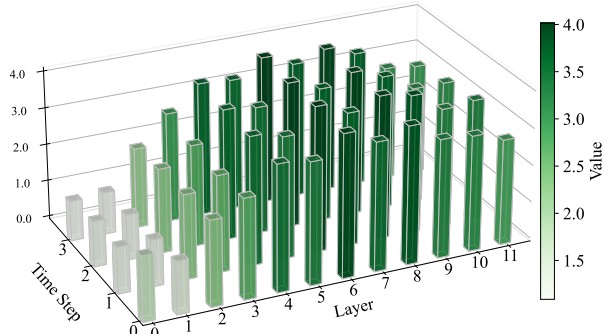

*Figure 6.* Distribution of the learnable parameter $\gamma$ in the 12-layer SpikingLM. The progressive growth of $\gamma$ along network depth mitigates gradient vanishing.

### 4.4. Effect of the Temporal Parameter in Spike2Max

To investigate the behavior of $\gamma$, we visualize its distribution throughout SpikingLM. As illustrated in Fig. 6, Spike2Max incorporates a learnable temporal parameter $\gamma$ to modulate the scale of the attention distribution. We observe that $\gamma$ tends to increase with network depth, which helps alleviate gradient vanishing in deeper layers. This layer-wise trend suggests that deeper spiking blocks require stronger temporal amplification to preserve informative spike signals. Moreover, timestep-specific $\gamma$ values allow Spike2Max to adjust the attention scale across temporal dynamics instead of applying a single global factor. By incorporating this temporal specificity, Spike2Max more effectively captures the inherent temporal dependencies of SNNs.

### 4.5. Comparison with Spiking Self-Attention

Spike2Max improves token selectivity by applying $2^{A-\max(A)-1}$ to the attention logits. Under this formulation,

*Table 3.* Perplexity comparison between vanilla SSA and Spike2Max in the same language backbone.

| Attention | PPL $\downarrow$ | Energy (mJ) $\downarrow$ |
|---|---|---|
| SSA (Zhou et al., 2022) | 24.58 | 1.580 |
| **Spike2Max** (Ours) | **19.78** | **1.457** |

the token with the largest logit receives weight $0.5$, while each one-unit decrease in the logit halves the corresponding weight. This exponential decay recovers the concentration behavior of softmax attention while retaining shift-based computation. To directly isolate this effect, we conduct a controlled comparison by replacing Spike2Max with SSA in the same backbone and training setup.

As shown in Tab. 3, Spike2Max improves perplexity by 4.8 while reducing energy by 7.8% compared with SSA. This result indicates that the exponential decay introduced by Spike2Max provides stronger key-token selectivity without increasing the hardware cost.

### 4.6. Underflow at Low Precision

We further evaluate whether the shift-based implementation of Spike2Max remains numerically reliable under low-precision registers. After max-subtraction, the Spike2Max factor $2^{z_i-\max(z)-1}$ is realized by integer right-shifts. If the required shift exceeds the available bit-width, the corresponding value underflows and is truncated to zero. To quantify this effect, we conduct a Monte Carlo simulation with sequence length $N = 64$ and key dimension $d_k = 64$. Following the spike coincidence statistics, logits are sampled from $\text{Binomial}(d_k, r^2)$ under firing rate $r$, and full-precision Spike2Max is used as the reference.

As shown in Tab. 4, 8-bit registers make the underflow error negligible across the tested firing rates. This behavior is expected: underflow only removes entries far below the row maximum, so the truncation acts as a hard top-$k$ effect on

*Table 4.* Underflow error of low-precision Spike2Max, measured by MSE against full-precision Spike2Max.

| Rate $r$ | 4-bit MSE | 6-bit MSE | 8-bit MSE |
|---|---|---|---|
| 0.05 | $3.6 \times 10^{-3}$ | $3 \times 10^{-6}$ | $\approx 0$ |
| 0.10 | $2.6 \times 10^{-2}$ | $4.7 \times 10^{-4}$ | $\approx 0$ |

already negligible attention weights. At inference time, the learnable temporal scale $\gamma_t$ is folded into the LIF threshold as $V'_{th} = V_{th}/\gamma_t$, leaving Spike2Max to use integer shifts and comparisons without floating-point intermediate values.

### 4.7. Energy Efficiency

*Table 5.* Energy Consumption and Perplexity Analysis

| Method | Energy(mJ) ↓ | PPL ↓ |
|---|---|---|
| *Overall Ablation* | | |
| + Distribution-aware Scaling | 1.545 | 24.47 |
| + Spike2Max | 1.485 | 256.39 |
| + Both | **1.457** | **19.78** |
| *Ablation for Distribution-aware Scaling* | | |
| + RMSNorm | 1.643 | 22.18 |
| + Fixed Scaling | **1.456** | 45.93 |
| + Distribution-aware Scaling (Ours) | 1.545 | **19.78** |
| *Ablation for Spike2Max* | | |
| + Standard Softmax | 1.761 | 18.29 |
| + Max-Sub | **1.350** | 22.75 |
| + Spike2Max (Ours) | 1.457 | **19.78** |

We estimate energy consumption using operation counts and costs from Horowitz (2014). Tab. 5 demonstrates that combining Distribution-aware Scaling and Spike2Max achieves the optimal performance-efficiency trade-off. Distribution-aware Scaling proves essential for convergence, preventing performance collapse, while outperforming RMSNorm with a 6.0% reduction in energy consumption. Complementing this, Spike2Max leverages shift operations to reduce energy consumption by 17.3% compared to Softmax. Despite an energy cost compared to Max-Sub, Spike2Max achieves gains via learnable temporal parameters, confirming the importance of temporal adaptability.

Fig. 7 illustrates the firing rates of LIF neurons across different blocks of SpikingLM. In the baseline model, the firing activity gradually decreases in deeper layers, indicating a severe dead-neuron issue and weakened information propagation. After incorporating Distribution-aware Scaling and Spike2Max, SpikingLM maintains more stable spike activity across layers, effectively alleviating the vanishing-spike problem in deep blocks. This suggests that the proposed modules improve the dynamic range of spiking representations and preserve useful token interactions throughout

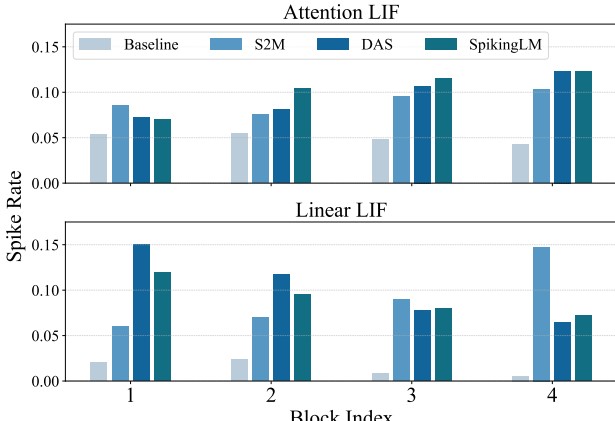

*Figure 7.* Estimated energy consumption per token. Attention LIF and Linear LIF denote the LIF neurons between the attention and linear layers, and at the block output, respectively.

the network. Moreover, SpikingLM avoids excessive firing activity, achieving a favorable trade-off between firing-rate-induced energy consumption and model accuracy.

## 5. Conclusion

In this paper, we presented SpikingLM, a novel framework designed to bridge the gap between energy-efficient SNNs and high-performance language modeling. We addressed the two fundamental challenges limiting SNNs in NLP: gradient degradation caused by dead neurons and the lack of token selectivity in attention mechanisms. To overcome the issues, we introduced Distribution-aware Scaling, a module that ensures gradient propagation and convergence. Crucially, Distribution-aware Scaling can be fused into linear layers during inference, incurring zero computational overhead. To resolve the latter, we proposed Spike2Max, a hardware-efficient mechanism that restores the critical winner-takes-all dynamics via integer-based bit-shifting operations. Empirical evaluations demonstrate that SpikingLM achieves performance competitive with conventional ANNs while significantly reducing energy consumption. This work paves the way for sustainable, large-scale AI deployment.

## Acknowledgements

This work was supported by the Fundamental and Inter-disciplinary Disciplines Breakthrough Plan of the Ministry of Education of China (JYB2025XDXM102), the National Natural Science Foundation of China (Grants 62576080 and 62220106008), the Guangdong Introducing Innovative and Entrepreneurial Teams (Grant 2023ZT10×044), and the Shenzhen Science and Technology Research Fund (Grant JCYJ20220818103001002).

## Impact Statement

This paper presents work whose goal is to advance the field of Machine Learning. There are many potential societal consequences of our work, none which we feel must be specifically highlighted here.

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

# A. Proof of Spike2Max Theorem

## A.1. Problem Setting

In SNNs, division operations are computationally expensive. We propose an unnormalized approximation to softmax attention:

$$\text{Spike2Max}(z) = \gamma_t \cdot 2^{z-\max(z)-1}, \tag{13}$$

where $\gamma_t > 0$ is a learnable, time-dependent scale factor that does **not** depend on the input $z$. This avoids the summation required for normalization.

## A.2. Main Theorem

**Theorem A.1** (Spike2Max Attention Output Approximation). *Let $z \in \mathbb{R}^n$ be attention logits with $m = \max_j z_j$. Define:*

- *Softmax weights: $p_i = \frac{e^{z_i}}{\sum_{j=1}^n e^{z_j}}$*

- *Spike2Max weights: $\hat{w}_i = \gamma_t \cdot 2^{z_i - m - 1}$*

*For any value matrix $V \in \mathbb{R}^{n \times d_v}$, the attention output error satisfies:*

$$\left\| p^\top V - \hat{w}^\top V \right\|_2 \leq \mathcal{E}(\gamma_t, z) \cdot \|V\|_F, \tag{14}$$

*where the approximation error factor is:*

$$\mathcal{E}(\gamma_t, z) = \sqrt{\sum_{i=1}^n \left( p_i - \gamma_t \cdot 2^{z_i - m - 1} \right)^2}. \tag{15}$$

*Moreover, there exists an optimal scale factor $\gamma_t^*$ that minimizes $\mathcal{E}$:*

$$\gamma_t^* = \frac{\sum_{i=1}^n p_i \cdot 2^{z_i - m - 1}}{\sum_{i=1}^n 2^{2(z_i - m - 1)}}, \tag{16}$$

*and the minimum error satisfies:*

$$\mathcal{E}^* \leq (1 - \ln 2) \cdot \sigma_p(z) + \mathcal{O}\left( (1 - \ln 2)^2 \right), \tag{17}$$

*where $\sigma_p(z) = \sqrt{\sum_i p_i (z_i - \mu_p)^2}$ is the softmax-weighted standard deviation of logits and $\mu_p = \sum_i p_i z_i$.*

## A.3. Proof

*Proof.* We prove the theorem in several steps.

**Step 1: Setup and Notation.** Let $\tilde{z}_i = z_i - m - 1 \leq -1$ denote the shifted logits. Define:

$$a_i = e^{\tilde{z}_i}, \quad Z_e = \sum_{j=1}^n a_j, \quad p_i = \frac{a_i}{Z_e}, \tag{18}$$

$$b_i = 2^{\tilde{z}_i} = e^{\tilde{z}_i \ln 2} = a_i^{\ln 2}. \tag{19}$$

The key relationship between $a_i$ and $b_i$ is:

$$b_i = a_i \cdot e^{-\delta \tilde{z}_i} = a_i \cdot a_i^{-\delta} = a_i^{1-\delta} = a_i^{\ln 2}, \tag{20}$$

where $\delta = 1 - \ln 2 \approx 0.307$.

**Step 2: Output Error Bound.** The attention outputs are:

$$o = p^\top V = \sum_{i=1}^{n} p_i v_i, \tag{21}$$

$$\hat{o} = \hat{w}^\top V = \sum_{i=1}^{n} \gamma_t b_i v_i. \tag{22}$$

By Cauchy-Schwarz inequality:

$$\|o - \hat{o}\|_2 = \left\|\sum_{i=1}^{n} (p_i - \gamma_t b_i) v_i\right\|_2 \tag{23}$$

$$\leq \sqrt{\sum_{i=1}^{n} (p_i - \gamma_t b_i)^2} \cdot \sqrt{\sum_{i=1}^{n} \|v_i\|_2^2} \tag{24}$$

$$= \mathcal{E}(\gamma_t, z) \cdot \|V\|_F. \tag{25}$$

**Step 3: Optimal Scale Factor.** To find $\gamma_t^*$, we minimize the squared error:

$$L(\gamma_t) = \sum_{i=1}^{n} (p_i - \gamma_t b_i)^2. \tag{26}$$

Taking the derivative and setting to zero:

$$\frac{\partial L}{\partial \gamma_t} = -2 \sum_{i=1}^{n} (p_i - \gamma_t b_i) b_i = 0, \tag{27}$$

$$\sum_{i=1}^{n} p_i b_i = \gamma_t \sum_{i=1}^{n} b_i^2. \tag{28}$$

Therefore:

$$\gamma_t^* = \frac{\sum_{i=1}^{n} p_i b_i}{\sum_{i=1}^{n} b_i^2} = \frac{\sum_{i=1}^{n} \frac{a_i}{Z_e} \cdot b_i}{\sum_{i=1}^{n} b_i^2} = \frac{\sum_{i=1}^{n} a_i b_i}{Z_e \sum_{i=1}^{n} b_i^2}. \tag{29}$$

**Step 4: Minimum Error Characterization.** Substituting $\gamma_t^*$ back:

$$\mathcal{E}^{*2} = \sum_{i=1}^{n} p_i^2 - 2\gamma_t^* \sum_{i=1}^{n} p_i b_i + (\gamma_t^*)^2 \sum_{i=1}^{n} b_i^2 \tag{30}$$

$$= \sum_{i=1}^{n} p_i^2 - \frac{\left(\sum_{i=1}^{n} p_i b_i\right)^2}{\sum_{i=1}^{n} b_i^2}. \tag{31}$$

This is the variance not explained by the linear fit of $p_i$ onto $b_i$.

**Step 5: First-Order Approximation Analysis.** To obtain an interpretable bound, we use the relationship $b_i = a_i \cdot e^{-\delta \tilde{z}_i}$ and expand for small $\delta$:

$$b_i = a_i(1 - \delta \tilde{z}_i + \mathcal{O}(\delta^2 \tilde{z}_i^2)). \tag{32}$$

Thus:

$$p_i - \gamma_t b_i = \frac{a_i}{Z_e} - \gamma_t a_i(1 - \delta \tilde{z}_i + \mathcal{O}(\delta^2)). \tag{33}$$

At the optimal $\gamma_t^* \approx \frac{1}{Z_e}$ (to first order), we have:

$$p_i - \gamma_t^* b_i \approx \frac{a_i}{Z_e} \cdot \delta \tilde{z}_i = p_i \cdot \delta \tilde{z}_i. \tag{34}$$

Therefore:

$$\mathcal{E}^* \approx \delta \sqrt{\sum_{i=1}^n p_i^2 \tilde{z}_i^2} \leq \delta \sqrt{\sum_{i=1}^n p_i \tilde{z}_i^2} = \delta \cdot \sigma_p(\tilde{z}), \tag{35}$$

where we used $p_i \leq 1$ and $\sigma_p(\tilde{z}) = \sqrt{\mathbb{E}_p[\tilde{z}^2] - (\mathbb{E}_p[\tilde{z}])^2}$.

Since $\tilde{z}_i = z_i - m - 1$, we have $\sigma_p(\tilde{z}) = \sigma_p(z)$. $\qquad\square$

## A.4. Corollaries for Practical Use

**Corollary A.2** (Learnable $\gamma_t$ Approximation). *When $\gamma_t$ is learned via gradient descent on attention output error, it converges to a value $\hat{\gamma}_t$ satisfying:*

$$\mathbb{E}_{z \sim \mathcal{D}} \left[ \mathcal{E}(\hat{\gamma}_t, z)^2 \right] \leq \mathbb{E}_{z \sim \mathcal{D}} \left[ \mathcal{E}(\gamma_t^*(z), z)^2 \right] + Var_{z \sim \mathcal{D}}[\gamma_t^*(z)] \cdot \mathbb{E}[Z_2^2], \tag{36}$$

*where $\mathcal{D}$ is the distribution of attention logits and $Z_2 = \sum_j 2^{\tilde{z}_j}$.*

*In practice, if the logit distribution is concentrated (low variance in $\gamma_t^*$ across samples), a single learned $\gamma_t$ achieves near-optimal approximation.*

**Corollary A.3** (Relative Attention Error). *Define the relative output error as:*

$$\mathcal{E}_{rel} = \frac{\|p^\top V - \hat{w}^\top V\|_2}{\|p^\top V\|_2}. \tag{37}$$

*For attention with concentrated weights (effective number of attended positions $n_{eff} = 1 / \sum_i p_i^2$), the relative error satisfies:*

$$\mathcal{E}_{rel} \leq \frac{(1 - \ln 2) \cdot \sigma_p(z)}{\sqrt{n_{eff}}} \cdot \frac{\|V\|_F}{\|p^\top V\|_2}. \tag{38}$$

## A.5. Bound for SNN Attention

**Proposition A.4** (SNN-Specific Bound). *For spiking attention with binary queries and keys $Q, K \in \{0, 1\}^{T \times d_k}$, the logits $z = QK^\top$ satisfy $z_i \in \{0, 1, \ldots, d_k\}$. The approximation error is bounded by:*

$$\mathcal{E}^* \leq (1 - \ln 2) \cdot \sqrt{d_k} \approx 0.307 \sqrt{d_k}. \tag{39}$$

*Proof.* To analyze the scale of the logits $z_i$, we consider the entries of $Q$ and $K$ as independent Bernoulli random variables with activation probability $\rho$. Each logit $z_i = \sum_{l=1}^{d_k} Q_{il} K_{jl}$ can be modeled as a sum of $d_k$ independent Bernoulli trials $X_l = Q_{il} K_{jl}$, where $P(X_l = 1) = \rho^2$. Under this statistical independence, the variance of the logits is given by:

$$\mathrm{Var}(z) = d_k \cdot \mathrm{Var}(X_l) = d_k \rho^2 (1 - \rho^2). \tag{40}$$

The standard deviation, which characterizes the effective range of fluctuations in the attention scores, scales as:

$$\sigma(z) = \sqrt{\rho^2 (1 - \rho^2)} \cdot \sqrt{d_k}. \tag{41}$$

For concentrated attention distributions $p$ typical in SNNs, the dispersion $\sigma_p(z)$ is dominated by this inherent statistical scale. Taking the maximum possible value of the term $\sqrt{\rho^2(1 - \rho^2)} = 0.5$ (at $\rho^2 = 0.5$) and incorporating the pre-defined approximation constant $(1 - \ln 2)$, we obtain:

$$\mathcal{E}^* \leq (1 - \ln 2) \cdot \sigma_p(z) \lesssim (1 - \ln 2) \sqrt{d_k}. \tag{42}$$

This confirms that the error bound grows with the square root of the head dimension $d_k$. $\qquad\square$

*Table 6.* Task-specific fine-tuning hyperparameter settings on the GLUE benchmark.

| Dataset | CoLA | SST-2 | MRPC | STS-B | QQP | MNLI | QNLI | RTE |
|---|---|---|---|---|---|---|---|---|
| Learning Rate | $5 \times 10^{-5}$ | $2 \times 10^{-5}$ | $2 \times 10^{-5}$ | $1 \times 10^{-4}$ | $5 \times 10^{-5}$ | $3 \times 10^{-5}$ | $5 \times 10^{-5}$ | $5 \times 10^{-6}$ |
| Weight Decay | 0.2 | 0.01 | 0 | 0 | 0.1 | 0.1 | 0.01 | 0.01 |
| Epochs | 20 | 20 | 20 | 20 | 10 | 10 | 10 | 20 |

## B. Implementation Details

### B.1. Hardware and Software Environment

All experiments were implemented using the PyTorch and SpikingJelly (Fang et al., 2023) framework. The pre-training and fine-tuning processes were conducted on a high-performance computing cluster equipped with $8\times$ NVIDIA H100 (80GB) GPUs. The software environment consisted of Ubuntu 22.04 LTS, Python 3.10, and the Hugging Face Transformers library.

### B.2. Model Architectures

We utilized two primary configurations for our experiments:

- **Main**: `bert-base-uncased` (12 layers, 768 hidden units, 12 attention heads).

- **Ablation**: `bert_uncased_L-4_H-768_A-12` (4 layers, 768 hidden units, 12 attention heads) was employed to ensure computational efficiency during extensive parameter studies.

### B.3. Main Pre-training Setup

**Configurations.** For our primary results, we pre-train the `bert-base-uncased` backbone. The model is trained for 800,000 steps with a global batch size of 64 and a maximum sequence length of 128. We adopt a learning rate of $2 \times 10^{-4}$ with 5,000 linear warmup steps.

**Distribution-aware Scaling Initialization.** In the final full-network training runs, we used a fixed integer approximation for the Distribution-aware Scaling scale initialization for implementation simplicity. Instead of recomputing the RMS-based estimate in Eq. 7 for every run, all learnable scaling vectors were initialized as $\lambda = 7 \cdot \mathbf{1}$. This value was chosen as a convenient integer approximation to the empirical RMS initialization scale observed in preliminary runs, after which $\lambda$ remained learnable and was updated by gradient descent.

**Data and Monitoring.** The model was pre-trained on a large-scale unlabeled English corpus, following the standard BERT pre-training setting. The dataset mixture includes: **STORIES**[1], **BookCorpus**[2], **CC-News**[3], **OpenWebText**[4], and **English Wikipedia**[5]. These datasets provide broad linguistic coverage across diverse domains. We adopted standard text preprocessing pipelines consistent with BERT to ensure data quality.

### B.4. Task-Specific Hyperparameter Settings

Table 6 provides the specific choices of learning rate ($\eta$), weight decay ($\lambda$), and training epochs ($E$) for each task in the GLUE benchmark. For all tasks, we consistently set the batch size to 32, the maximum sequence length to 128, and the number of warmup steps to 0. These configurations were optimized to ensure stable convergence and fair comparison across different linguistic domains.

### B.5. Ablation Study Configurations

**Pre-training Setup.** We utilize the `bert_uncased_L-4_H-768_A-12` architecture for all ablation studies. The model is pre-trained for 500,000 steps with a batch size of 64 and a maximum sequence length of 128. We employ the AdamW

---

[1] https://huggingface.co/datasets/lucadiliello/STORIES
[2] https://huggingface.co/datasets/bookcorpus/bookcorpus
[3] https://huggingface.co/datasets/vblagoje/cc_news
[4] https://huggingface.co/datasets/Skylion007/openwebtext
[5] https://huggingface.co/datasets/wikimedia/wikipedia

optimizer with a learning rate of $2 \times 10^{-4}$ and 5,000 warmup steps.

**Downstream Fine-tuning.** We evaluate the pre-trained variants on the GLUE benchmark. The task-specific hyperparameter configurations, including learning rates and training epochs for CoLA, SST-2, MRPC, QNLI, RTE, STS-B, QQP, and MNLI, are summarized in Table 7.

*Table 7.* Hyperparameter settings for GLUE tasks (Ablation Study).

|  | Batch Size | Learning Rate | Epochs | Max Length | Optimizer | Weight Decay | Warmup Ratio |
|---|---|---|---|---|---|---|---|
| **Hyperparameter** | 32 | $2 \times 10^{-4}$ | 5 | 128 | AdamW | 0.01 | 0.1 |

## C. Energy Consumption Analysis

To quantify the efficiency of our Spiking Neural Network (SNN) approach, we estimate the energy consumption based on synaptic operations (SOPs).

### C.1. Energy Estimation Methodology

Following the standard SNN energy model, the total energy $E_{total}$ is calculated as (Wei et al., 2025; Zhang et al., 2024; Liu et al., 2026):

$$E_{total} = \sum_{t=1}^{T} \left( \sum_{l=1}^{L} E_{SOP} \cdot \text{SOP}_l^t + E_{FLOP} \cdot \text{FLOP}_{l,\text{static}} \right), \tag{43}$$

where $T$ is the number of time steps, $E_{SOP}$ is the energy required for a spiking addition operation, and $E_{FLOP}$ represents the energy for floating-point operations in non-spiking layers (e.g., embeddings). We use the reference values from (Horowitz, 2014) assuming $E_{add} = 0.9$ pJ and $E_{mac} = 3.7$ pJ for 32-bit operations in 45nm technology.

## D. More Experiment Results

**Firing rate** As illustrated in Fig. 8, the firing rates of various components in the SpikingLM exhibit distinct layer-wise dynamics. High-activity modules, specifically Query and Key, maintain relatively elevated firing rates in shallower layers before undergoing a fluctuating decline as the model depth increases. The convergence of these firing rates across different modules in deeper layers demonstrates that the SpikingLM effectively ensures the optimal firing rates throughout the network.

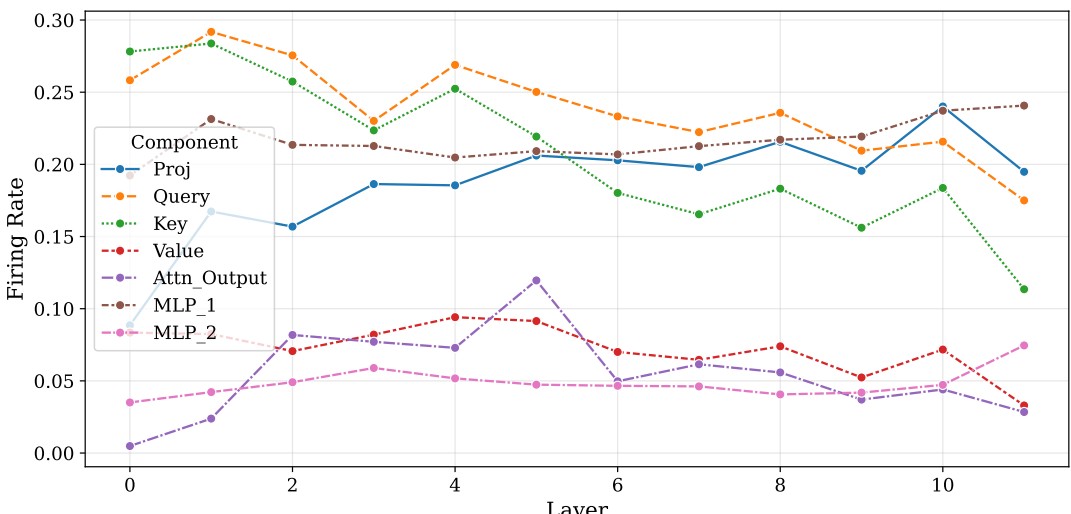

*Figure 8.* Layer-wise firing rates of different components in the Spiking BERT model.

**Weight distribution** As shown in the Fig. 9, the weight distributions across all components remain stable and concentrated near zero throughout all layers. This consistent distribution prevents numerical instability during training, indicating that SpikingLM effectively ensures the optimal firing rates of the network.

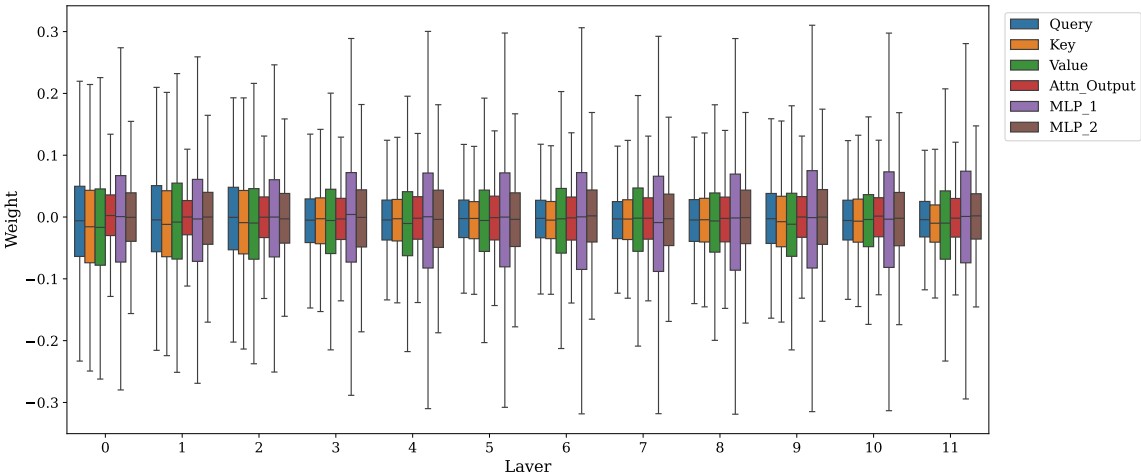

*Figure 9.* Layer-wise weight distributions across different model components.

**LIF neuron input distributions** We visualized the LIF neuron input distributions across all layers (from layer 0 to 11), divided into three stages as shown in Fig. 10 (Layers 0-3), Fig. 11 (Layers 4-7), and Fig. 12 (Layers 8-11). The distributions across key components like Query, Key, and MLP remain consistently centered and bounded.

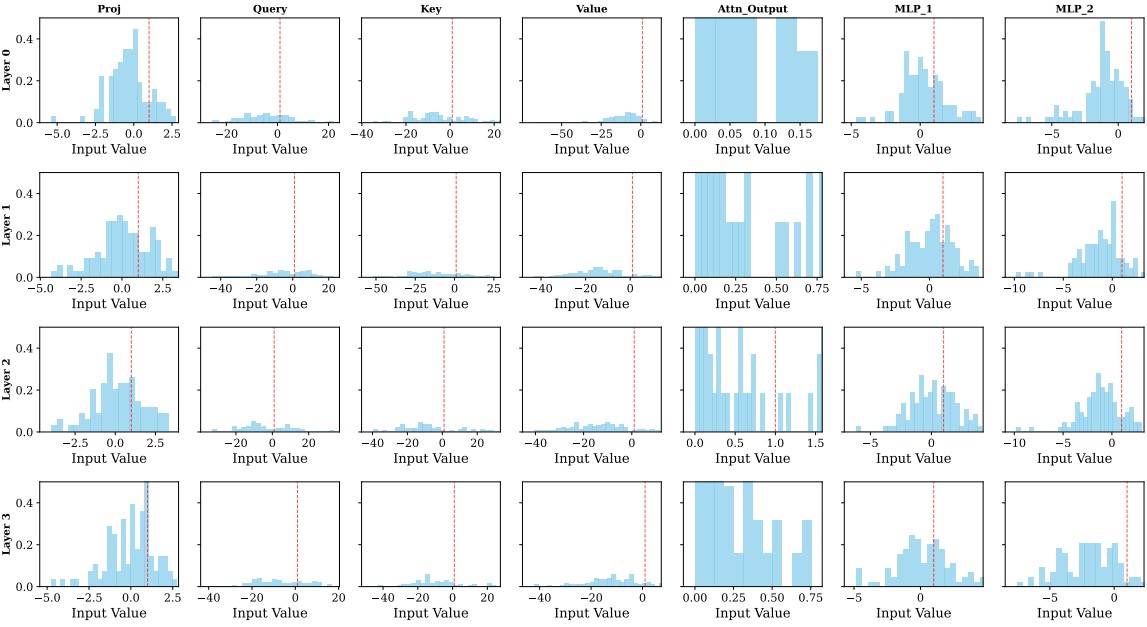

*Figure 10.* LIF neuron input distributions for Layers 0-3.

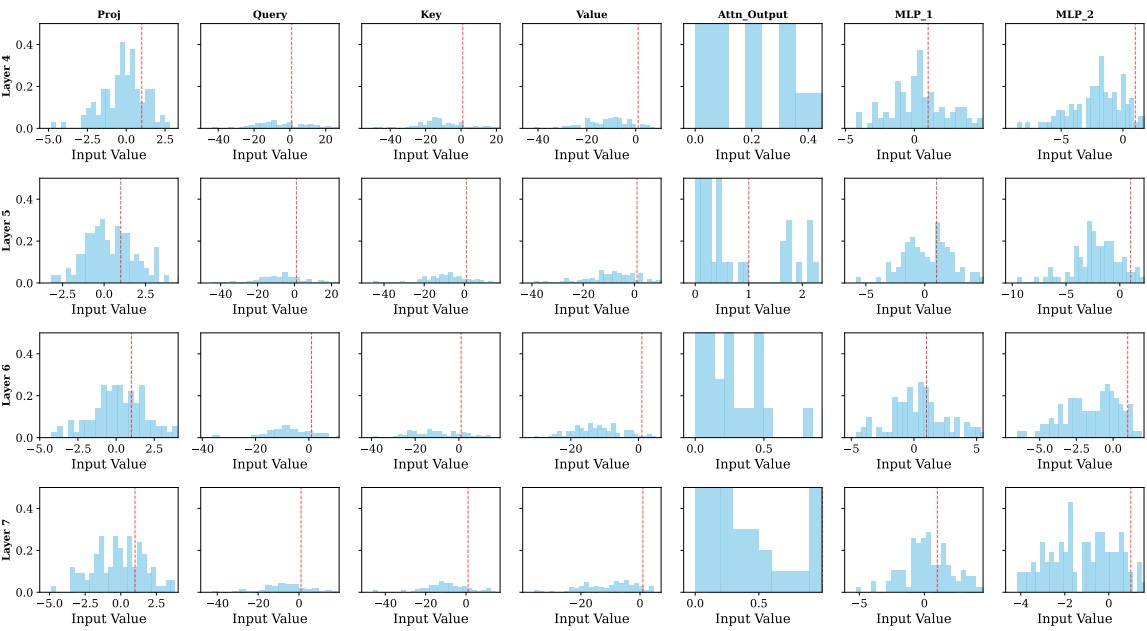

*Figure 11.* LIF neuron input distributions for Layers 4-7.

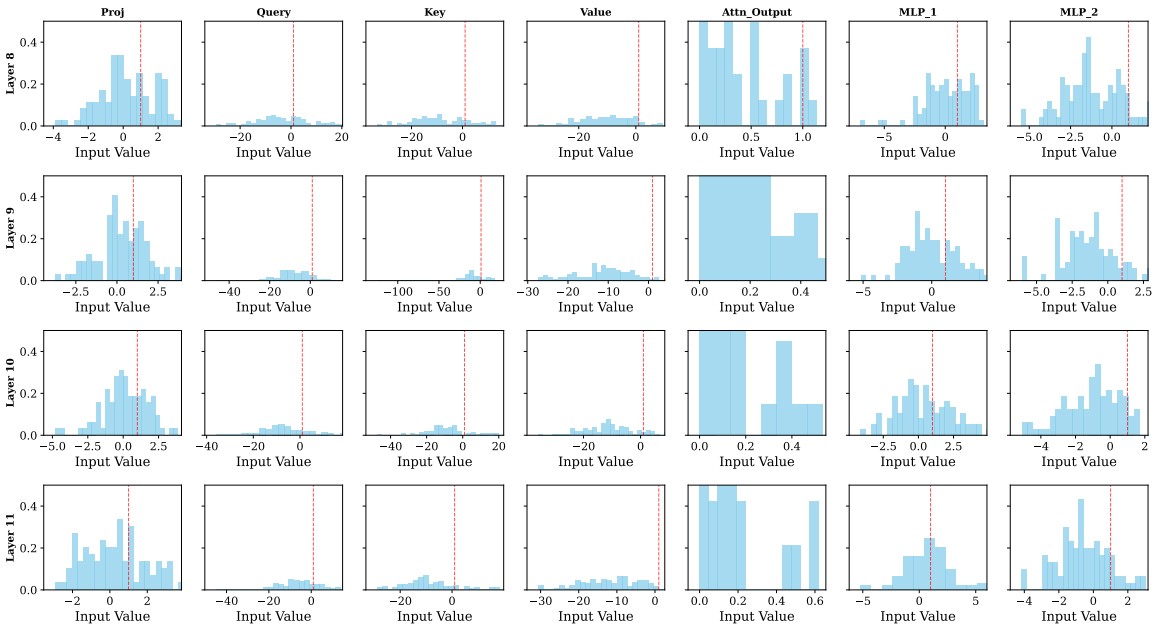

*Figure 12.* LIF neuron input distributions for Layers 8-11.

