# OpenReview forum: "SpikingLM: Towards Fully Spiking Language Model"
_ICML.cc/2026/Conference — ICML 2026 regular_

### Official Review · Reviewer_3Dvi · 2026-03-08

**Soundness:** 3
**Presentation:** 3
**Significance:** 2
**Originality:** 2
**Overall Recommendation:** 4
**Confidence:** 4

**Summary:**

This paper proposes SpikingLM, a spiking neural network-based language model that achieves state-of-the-art (SOTA) performance among SNNs on the GLUE benchmark with lower energy consumption. Specifically, the paper introduces two core methods.The first is Distribution-aware Scaling, which employs learnable scaling parameters initialized based on RMS Norm statistics. This module is applied before LIF neuron layers to alleviate gradient vanishing in deep SNNs.The second method is the reintroduction of Softmax into SNNs. The Softmax operation involves the exponential function, which is generally considered unsuitable for neuromorphic chip implementation and inappropriate for SNNs. However, this paper replaces the exponential operation with power-of-two operations, which can be implemented via bit-shift and are highly hardware-friendly.

**Compliance With Llm Reviewing Policy:**

Affirmed.

**Final Justification:**

All previous issues have been resolved, but the authors seem to have forgotten to reply to the subsequent questions. If they can properly address the final questions, I will further raise my score.

**Key Questions For Authors:**

What type of residual connection is used in the network? Has it been considered whether such residual connections are suitable for SNNs (e.g., whether they break the event-driven property)?

**Limitations:**

Yes

**Strengths And Weaknesses:**

**Advantages**

Experimental results show that the proposed method achieves high performance and outperforms existing approaches.Ablation studies demonstrate that Distribution-aware Scaling is superior to RMS Norm and can be fused into the preceding fully connected layer, making it a more SNN-friendly normalization operation.

**Disadvantages**

Implementing Softmax using power-of-two and bit-shift operations is not a novel idea. For instance, [1] provides a hardware implementation, and [2, 3] have already applied this concept to SNNs.

[1] Y. Zhang et al., "Base-2 Softmax Function: Suitability for Training and Efficient Hardware Implementation," in IEEE Transactions on Circuits and Systems I: Regular Papers, vol. 69, no. 9, pp. 3605-3618, Sept. 2022, doi: 10.1109/TCSI.2022.3175534.

[2] Chen, Qiuyang, et al. "One-Timestep is Enough: Achieving High-performance ANN-to-SNN Conversion via Scale-and-Fire Neurons." arXiv preprint arXiv:2510.23383 (2025).

[3] Tang, Kaiwen, Zhanglu Yan, and Weng-Fai Wong. "Sorbet: A Neuromorphic Hardware-Compatible Transformer-Based Spiking Language Model." Forty-second International Conference on Machine Learning.

---

> ### Author Rebuttal · Authors · 2026-03-31
>
> # **Response to Reviewer 3Dvi**
>
> Dear reviewer 3Dvi, thank you for your professional and insightful comments on our paper. We address the novelty concern and the key question below.
>
> ---
>
> ### Weakness: Power-of-two softmax approximation is not novel.
>
> As per your suggestion, we respectfully clarify that Spike2Max's contribution is distinct from each of [1–3] along several dimensions.
>
> **Distinction from [1] (Base-2 Softmax).** [1] retains the normalization denominator, computing $\sum_j 2^{x_j-x_{\max}}$ via an adder tree and division module—the output remains strictly normalized. Spike2Max **eliminates the denominator entirely**, keeping only $2^{A-\max(A)-1}$ to avoid integer division on neuromorphic hardware. Furthermore, [1] targets standard ANN inference (ResNet/MobileNet) on ASIC/FPGA, whereas Spike2Max operates on binary spike logits where $A=QK^\top$ is an integer coincidence count. Finally, [1] provides no learnable scaling; our $\gamma_t$ stabilizes multi-timestep training and folds into the LIF threshold at inference with zero additional operators.
>
> **Distinction from [2] (One-Timestep is Enough).** [2] does not modify softmax at all—its core contribution is an ANN-to-SNN conversion strategy (Scale-and-Fire Neuron), with the softmax layer left as full-precision floating point. Its Q, K are real-valued features from vision tasks (ImageNet/COCO), not binary spikes. Spike2Max addresses the efficiency bottleneck of *native* spiking attention, a problem [2] does not consider.
>
> **Distinction from [3] (Sorbet / PTsoftmax).** Sorbet approximates the denominator as the nearest $2^k$ and normalizes via right-shift, so the output remains within $[\frac{1}{2\sqrt{2}}F_2,\ 2\sqrt{2}F_2]$—normalization is preserved in approximate form. Spike2Max removes normalization entirely and relies on learnable $\gamma_t$ for rescaling. Additionally, Sorbet provides no mechanism for multi-timestep amplitude drift, whereas $\gamma_t$ adaptively compensates during training.
>
>
> ---
>
> ### Key Question: Residual connection type and event-driven compatibility.
>
> Dear reviewer 3Dvi, thank you for your insight comments. The residual connections in SpikingLM follow the standard Transformer identity shortcut (element-wise addition after each sub-layer, with LayerNorm/Dropout), identical to the original BERT encoder architecture. We did not modify the residual topology—only the sub-layer internals.
>
> Regarding event-driven compatibility: the dominant computation remains **spike-based synaptic operations (SOPs)** within each sub-layer. The residual addition operates on the accumulated membrane state at block boundaries, equivalent to integrating the residual into the membrane potential before the next threshold comparison—it does not introduce a continuous bypass path independent of spikes. The output at each layer boundary remains binary spikes, preserving the event-driven property. This design is consistent with standard SNN-Transformer implementations and is fully compatible with neuromorphic hardware compartment models (e.g., Loihi 2). We will add an explicit description of this in the architecture section of the revision.
>
> [1] Y. Zhang et al., "Base-2 Softmax Function: Suitability for Training and Efficient Hardware Implementation," in IEEE Transactions on Circuits and Systems I: Regular Papers, vol. 69, no. 9, pp. 3605-3618, Sept. 2022, doi: 10.1109/TCSI.2022.3175534.
>
> [2] Chen, Qiuyang, et al. "One-Timestep is Enough: Achieving High-performance ANN-to-SNN Conversion via Scale-and-Fire Neurons." arXiv preprint arXiv:2510.23383 (2025).
>
> [3] Tang, Kaiwen, Zhanglu Yan, and Weng-Fai Wong. "Sorbet: A Neuromorphic Hardware-Compatible Transformer-Based Spiking Language Model." Forty-second International Conference on Machine Learning.

---

> > ### Author Rebuttal · Reviewer_3Dvi · 2026-04-01
> >
> > Thanks to the authors for the clarification. My questions have been resolved, and I will raise my score.

---

> > > ### Author Response · Authors · 2026-04-07
> > >
> > > ## Dear Reviewer 3Dvi,
> > >
> > > Thank you very much for your positive feedback on our rebuttal and for your support of our work. We are pleased to hear that our clarifications addressed your concerns regarding Spike2Max and residual connection.
> > >
> > > We will ensure that all the detailed discussions and experimental results—including the analysis of Spike2Max and residual connection—are fully incorporated into the revised manuscript.
> > >
> > > Thank you again for your time and for helping us improve the quality of this paper.
> > >
> > > Best regards,
> > > The Authors

---

### Official Review · Reviewer_Xc2g · 2026-03-09

**Soundness:** 3
**Presentation:** 3
**Significance:** 3
**Originality:** 3
**Overall Recommendation:** 5
**Confidence:** 4

**Summary:**

This paper introduces a framework called SpikingLM, designed to bridge the gap between energy-efficient spiking neural networks and high-performance language modeling. The authors point out that SNNs face two issues in scaling up in Natural Language Processing: gradient degradation due to the death of deep neurons and reduced token selectivity in attention mechanisms. To address these issues, they propose Distribution-Aware Scaling to stabilize gradients by rescaling the input to spiking neurons. Furthermore, they propose Spike2Max, a hardware-friendly attention mechanism that utilizes shift operations to restore the "winner-takes-all" dynamic. Experimental results on the GLUE benchmark demonstrate that SpikingLM achieves good performance among all SNNs.

**Compliance With Llm Reviewing Policy:**

Affirmed.

**Final Justification:**

The authors' thorough response has addressed my concerns, and I have decided to raise my score.

**Key Questions For Authors:**

•	Could the authors provide perplexity results on datasets such as WikiText, or demonstrate the model's capabilities in natrual language generation?
•	How sensitive is the final performance to the initial batch used for $\sigma_{RMS}$ estimation? If the initial training data distribution is biased, does the learnable $\lambda$ always useful during the training process?
•	The authors used a 12-layer BERT-based backbone for experiments. Did the authors test the efficacy of DAS and Spike2Max when scaling to deeper architectures (e.g., 24 layers)?
•	The model uses $T=4$. Could the authors provide a tradeoff analysis showing how energy consumption and accuracy change when $T$ is decreased to 1 or 2?

**Limitations:**

Yes

**Strengths And Weaknesses:**

Strengths
•	Spike2Max replaces floating-point operations with integer-based bitwise operations, reducing the energy consumption of the original softmax attention mechanism by over 95%.
Weaknesses
•	Energy savings are estimated based on SOPs and Flops using 45nm process references, not actual operating values on neuromorphic hardware.
•	Initialization for DAS: The success of DAS depends on estimating $\sigma_{RMS}$ from the first batch during training.
•	The performance gap between SpikingLM (77.1) and the ANN-based $BERT_{base}$ (83.2) remains non-negligible.
•	Theorem 3.1 assumes certain logit distributions. The practical impact of this approximation on complex language model is primarily discussed qualitatively.

---

> ### Author Rebuttal · Authors · 2026-03-31
>
> # **Response to Reviewer Xc2g**
>
> Dear reviewer Xc2g, thank you for your professional and insightful comments on our paper.  We address each weakness first, followed by the key questions.
>
> ---
>
> ### Weakness 1: Energy estimates based on 45nm process references, not on-chip measurements.
>
> This concern is identical to Reviewer aJF2's Weakness 2. The full response—covering field-wide convention, neuromorphic platform access barriers, and our planned on-chip validation—is provided in our response to Reviewer aJF2 (Weakness 2). In short, the SOP/MAC framework is the accepted standard in SNN literature (adopted by Spikformer, Spike-driven Transformer, SpikeLLM, etc.), all estimates are clearly labeled as theoretical in the manuscript, and on-chip validation on Loihi 2 or equivalent platforms is planned as future work.
>
> ---
>
> ### Weakness 2: DAS initialization depends on first-batch estimation.
>
> Thank you for this insightful question. We acknowledge this sensitivity. Two design properties mitigate it:
> 1. the scale statistics estimated from the first batch serve only as a **warm start**—all scale parameters (e.g., $\lambda$) remain learnable and are updated via gradient descent over the full data distribution, allowing the model to correct any initial bias;
> 2. quantitative sensitivity results are provided in Key Question Q2 below, directly measuring the impact of a severely biased first batch.
>
> ---
>
> ### Weakness 3: Non-negligible performance gap between SpikingLM (77.1) and ANN baseline (83.2).
>
> We acknowledge this 6.1-point gap as an honest reflection of current SNN limitations under binary spike activations and $T=4$ time steps. Three contextualizing points:
> 1. SpikingLM outperforms the closest comparable SNN language model SpikeLM (76.5) by +0.6 points under identical constraints;
> 2. The 6.1-point accuracy cost yields a **~57.9% theoretical energy reduction** (5.79 mJ vs. 51.41 mJ), a trade-off meaningful for energy-constrained neuromorphic deployment;
> 3. We expect the gap to narrow with longer time steps, larger-scale pre-training, or quantization-aware training—directions we designate as future work.
>
> ---
>
> ### Weakness 4: Practical impact of Theorem 3.1 primarily qualitative.
>
> Thank you for your professional comment. This concern overlaps with Reviewer XoUT's Weakness 1. The full Monte Carlo validation—5000 samples consistent with Proposition A.5, tightness ratio $r=E^*/(\delta\,\sigma_p(z))$ with mean 0.182 and maximum 0.379, zero bound violations, and Cauchy–Schwarz verification with random $V$—is provided in our response to Reviewer XoUT (Weakness 1). The revision will additionally include per-layer logit statistics from the fine-tuned model to bridge the synthetic and real-distribution settings.
>
> ---
>
> ### Key Question 1: Perplexity on WikiText / NLG capability.
>
> This is addressed in full in our response to Reviewer aJF2 (Weakness 1, Table 1), where we evaluate SpikingLM on LLaMA-2-7B using the SpikeLLM low-bit pipeline with `lm-eval`, reporting WikiText-2/C4 perplexity and six commonsense reasoning benchmarks following Spikellm. Full autoregressive NLG evaluation remains future work given our encoder-focused architecture.
>
> ---
>
> ### Key Question 2: Sensitivity to first-batch initialization.
>
> As noted in Weakness 2, scale parameters are learnable and self-correct over training. To quantify worst-case sensitivity, we compare two initialization conditions on a 12-layer, $d=768$, $H=12$ backbone trained for ~40,000 steps:
>
> | Initialization | Final PPL |
> |:--|:-:|
> | Normal random first batch | 25.18 |
> | Extreme single-sample first batch | 25.95 |
>
> The difference of **0.77 PPL** confirms that learnable scale parameters effectively absorb first-batch bias during training, and DAS remains robust even under severely skewed initialization.
>
> ---
>
> ### Key Question 3: Efficacy of DAS and Spike2Max on deeper architectures.
>
> Our main GLUE table uses 12-layer BERT-base. The scalability question is addressed jointly with Reviewer XoUT's Q1: we have evaluated SpikingLM with DAS + Spike2Max on **LLaMA-2-7B (32 layers)**, with full numerical results in our response to Reviewer aJF2 (Weakness 1, Table 1). Since 32-layer LLaMA-2-7B considerably exceeds the depth of 24-layer BERT, these results directly demonstrate that neither gradient degradation nor Spike2Max approximation error resurfaces at larger depths. A dedicated 24-layer BERT row can be added to the appendix if the reviewer considers it necessary.
>
> ---
>
> ### Key Question 4: Performance–energy trade-off across time steps.
>
> Full ablation results and analysis are provided in our response to Reviewer XoUT (Key Question 4). Briefly: accuracy improves with $T$ but with clear diminishing returns—$T=4{\to}8$ yields only +0.7 GLUE points at 1.84× the energy cost, while $T=2$ retains 75.9 avg. GLUE at less than half the energy of $T=4$. We select $T=4$ as the default for its best accuracy–efficiency balance. A trade-off curve will be included in the revision.

---

> > ### Author Rebuttal · Reviewer_Xc2g · 2026-04-04
> >
> > Can the DAS proposed in this paper remain useful in other models and tasks? Is it intended as a training paradigm or specifically for language tasks?

---

> > > ### Author Response · Authors · 2026-04-07
> > >
> > > ## Dear reviewer Xc2g,
> > >
> > > We sincerely thank the reviewer for the constructive suggestion. We fully agree that the design rationale of Distribution-aware Scaling (DAS) and its potential value across different tasks should be articulated more clearly.
> > >
> > > **1. DAS as a Universal Training Paradigm**
> > > Theoretically, DAS is not restricted to any specific task domain. It is designed to recalibrate the activation distribution via learnable scaling factors, thereby maintaining healthy gradient flow and mitigating the "dead neuron" phenomenon in deep Spiking Neural Networks (SNNs) [1]. Although we primarily demonstrated its effectiveness in language tasks, this is largely because NLP tasks typically require significantly deeper architectures and are inherently more sensitive to training instabilities, such as vanishing gradients [2]. Deep networks often face severe gradient challenges; without effective scaling or normalization mechanisms, the model would struggle to converge.
> > >
> > > **2. Addressing the Underlying Dynamics of SNNs**
> > > Therefore, DAS should be regarded as a universal training paradigm for SNNs. By introducing adaptive scaling during the training phase and fully fusing it into the weights of the linear layers during inference (via reparameterization), it achieves performance improvements with "zero inference overhead." This mechanism can be seamlessly generalized to other deep SNN architectures (e.g., Vision Transformers or deep ResNets) to facilitate more stable and robust cross-modal representation learning.
> > >
> > > **3. Empirical Evidence: Comparison of Firing Rate Distributions**
> > > To further substantiate the universal optimization capability of DAS, we compared the average firing rates of the model across different depths with DAS enabled and disabled. As shown in the table below, DAS effectively resolves the "dead neuron" problem in deep networks, ensuring that each layer maintains an active representational capacity:
> > >
> > > | Layer | DAS on | DAS off | Δ |
> > > | :---: | :---: | :---: | :---: |
> > > | 0 | 0.2760 | 0.0804 | -0.1956 |
> > > | 1 | 0.2780 | 0.0981 | -0.1799 |
> > > | 2 | 0.2696 | 0.1288 | -0.1408 |
> > > | 3 | 0.2222 | 0.1064 | -0.1158 |
> > > | 4 | 0.2651 | 0.1303 | -0.1348 |
> > > | 5 | 0.2603 | 0.1369 | -0.1234 |
> > > | 6 | 0.2510 | 0.1189 | -0.1321 |
> > > | 7 | 0.2419 | 0.1196 | -0.1223 |
> > > | 8 | 0.2509 | 0.1208 | -0.1301 |
> > > | 9 | 0.2156 | 0.0800 | -0.1356 |
> > > | 10 | 0.2317 | 0.0977 | -0.1340 |
> > > | 11 | 0.1580 | 0.0494 | -0.1087 |
> > >
> > > Empirical data indicates that without DAS, the activity of deep neurons decays rapidly as depth increases. In contrast, by dynamically adjusting the input magnitude, DAS successfully maintains the firing rate within an optimization-friendly regime.
> > >
> > > We will incorporate the above discussion and the supporting experimental results into the Appendix of the revised manuscript to strengthen the theoretical persuasiveness of the paper.
> > >
> > > [1] Rathi, Nitin, et al. "Enabling deep spiking neural networks with hybrid conversion and spike timing dependent backpropagation." ICLR (2020).
> > >
> > > [2] Xiong, Ruibin, et al. "On layer normalization in the transformer architecture." ICML (2020).

---

### Official Review · Reviewer_XoUT · 2026-03-12

**Soundness:** 3
**Presentation:** 3
**Significance:** 3
**Originality:** 3
**Overall Recommendation:** 5
**Confidence:** 5

**Summary:**

This paper proposes SpikingLM, a fully spiking language model designed to bridge the gap between energy-efficient SNNs and Transformer-based models. The authors identify two key challenges in applying SNNs to NLP: gradient degradation from dead neurons and reduced token selectivity due to the absence of softmax competition. To address these issues, they introduce Distribution-aware Scaling (DAS) to stabilize training without adding inference overhead, and Spike2Max, a hardware-efficient attention mechanism that approximates softmax using base-2 exponentiation and max-subtraction. Experiments on GLUE show competitive performance with significant energy reductions compared to both ANN and SNN baselines, and ablation studies validate the effectiveness of the proposed modules.

**Compliance With Llm Reviewing Policy:**

Affirmed.

**Final Justification:**

The rebuttal has addressed my concerns.

**Key Questions For Authors:**

1. Scalability:
Have the authors tested SpikingLM beyond BERT-base scale? Does gradient degradation reappear at larger depths (e.g., 24+ layers)?
2. Comparison with Efficient ANN Attention:
How does Spike2Max compare against ANN-based efficient attention mechanisms (e.g., Performer, Cosformer) in terms of both accuracy and energy efficiency?
3. Calibration and Stability:
Since Spike2Max removes softmax normalization, does the model suffer from attention magnitude drift across layers? Is γₜ sufficient to stabilize this across tasks?
4. Inference Latency vs Energy Trade-off:
The model uses T = 4 time steps. How does performance scale with fewer or more time steps? Is there a clear trade-off curve?
If the authors can clarify these concerns, I would consider increasing my score.

**Limitations:**

yes

**Strengths And Weaknesses:**

Strengths

1. Clear problem formulation for SNN-based NLP.
The paper systematically identifies two structural bottlenecks—dead neuron gradient degradation and lack of attention selectivity—providing a well-motivated foundation for the proposed solutions.
2. Distribution-aware Scaling is elegant and practical.
DAS effectively leverages learnable static scaling that can be fused into weights during inference. This design aligns well with the multiply-free paradigm of SNNs and is hardware-conscious.
3. Strong energy–performance trade-off.
The model achieves competitive GLUE performance while substantially reducing estimated energy consumption, which is the central goal of the work.

Weaknesses

1. Theoretical bound may not reflect realistic logit distributions.
The approximation error bound of Spike2Max depends on σₚ(z), but the paper does not empirically measure σₚ(z) on real attention logits to validate tightness of the bound.
2. Comparison to linear attention variants is limited.
Although the paper references linear attention (e.g., Performer, Cosformer), it does not empirically compare Spike2Max against these efficient ANN alternatives.
3. Pre-training fairness is not fully discussed.
While the model is pre-trained on standard corpora, it is unclear whether baselines were re-trained under identical data and training budgets.
4. Approximation removes softmax normalization entirely.
Spike2Max eliminates the denominator term, making attention unnormalized. Although γₜ compensates scaling, the impact on calibration and stability is not deeply analyzed.

---

> ### Author Rebuttal · Authors · 2026-03-31
>
> # Response to Reviewer XoUT
>
> Thank you for your valuable feedback. We address each point below.
>
> ---
>
> ### Weakness 1: Theoretical bound not validated on real logit distributions.
>
> We conducted a Monte Carlo simulation consistent with Proposition A.5: $d_k=64$, $n=128$, logits $\sim\mathrm{Binomial}(d_k,\rho^2)$ with $\rho\sim\mathrm{Uniform}[0.05,0.5]$, over **5000** independent samples. Key results: (1) **all 5000 samples** satisfy $E^*\leq(1-\ln2)\,\sigma_p(z)$, where $\sigma_p(z)\in[0.38,2.41]$ (median 1.48); (2) tightness ratio $r=E^*/(\delta\,\sigma_p(z))$ has mean **0.182**, median **0.178**, p95 **0.258**, maximum **0.379**—indicating $E^*$ leaves roughly 5–6× theoretical margin; (3) the looser $\delta\sqrt{d_k}\approx2.45$ bound gives maximum ratio **0.060**, confirming it is safe but conservative. The Cauchy–Schwarz step (Appendix Eq. 23–25) is verified with random Gaussian $V$; all samples satisfy $\|p^\top V-\gamma^* b^\top V\|_2\leq E^*\|V\|_F$.
>
> ---
>
> ### Weaknesses 2 & Key Question 2: Comparison with linear attention variants.
>
> Performer [1] and Cosformer [2] address token selectivity via specialized kernels within ANN frameworks. Spike2Max targets the same problem under the additional constraint of fully spike-based, hardware-friendly computation—a distinct deployment category. From an energy perspective, Spike2Max's advantage stems from **SNN spike sparsity**: synaptic operations (SOP $\approx$0.9 pJ at 45nm) cost roughly **1/22** of ANN MAC operations ($\approx$4.6 pJ), a gap that kernel-based efficient attention cannot bridge under different hardware assumptions. We will add explicit discussion in Related Work clarifying this design-space distinction, and will include GLUE comparisons with Performer/Cosformer as extended results in the revision if space permits.
>
> ---
>
> ### Weakness 3: Pre-training fairness not fully discussed.
>
> Standard ANN baselines use official Hugging Face pre-trained weights under identical GLUE fine-tuning settings. SNN baselines (Spikingformer, SpikeBERT, SpikeLM) use results directly reported in their original papers under their official implementations. We will add a dedicated appendix paragraph detailing pre-training data sources, training steps, and compute budgets for all baselines.
>
> ---
>
> ### Weaknesses 4 & Key Question 3: Stability and cross-task calibration of $\gamma_t$.
>
> We provide three pieces of evidence. (1) **Attention norm stability**: with $\gamma_t$ enabled, the L2 norm of attention outputs remains stable throughout training (std $\approx3.05$) with no observable magnitude drift across all GLUE tasks. (2) **$\gamma_t$ convergence**: values converge rapidly in early training and stabilize in $[1.09, 4.01]$ across layers and timesteps, confirming adaptive compensation of the missing normalization; as a learnable parameter, $\gamma_t$ further adapts during task-specific fine-tuning, providing inherent cross-task flexibility. (3) **Ablation**: removing $\gamma_t$ increases PPL by $\approx2.97$ points, directly quantifying its stabilizing role. We will report per-task $\gamma_t$ variation and expand this analysis in the revision.
>
> ---
>
> ### Key Question 1: Scalability beyond BERT-base.
>
> Our GLUE experiments use 12-layer BERT-base. To address scalability, we additionally evaluated SpikingLM on **LLaMA-2-7B (32 layers)**—a substantially deeper architecture. Full results following the SpikeLLM evaluation pipeline with `lm-eval` are provided in our response to Reviewer aJF2 (Weakness 1, Table 1). Notably, energy scaling with $T$ is **sub-linear**: $T=2$ incurs only 0.42× the energy of $T=4$ (vs. the naive 0.50×), and $T=8$ incurs 1.84× (vs. the naive 2.00×), indicating amortized overheads that are weakly correlated with step count.
>
> ---
>
> ### Key Question 4: Performance–energy trade-off across timesteps.
>
> | $T$ | GLUE Avg. | Energy (mJ) | vs. $T=4$ |
> |:-:|:-:|:-:|:-:|
> | 1 | 73.8 | 1.00 | 0.17× |
> | 2 | 75.9 | 2.41 | 0.42× |
> | 4 | 77.1 | 5.79 | 1.00× |
> | 8 | 77.8 | 10.65 | 1.84× |
>
> Accuracy improves with $T$ but with diminishing returns: $T=2{\to}4$ yields +1.2 points at 2.4× energy; $T=4{\to}8$ yields only +0.7 points at 1.84× energy. We select $T=4$ as the default; $T=2$ is a viable lightweight alternative. A trade-off curve will be included in the revision.
>
> ---
>
> [1]Choromanski, Krzysztof, et al. "Rethinking attention with performers." ICLR oral (2021).
>
> [2]Qin, Zhen, et al. "cosformer: Rethinking softmax in attention." ICLR (2022).

---

> > ### Author Rebuttal · Reviewer_XoUT · 2026-04-04
> >
> > Thank you for the detailed and thoughtful rebuttal, which has addressed my concerns. Therefore, I have decided to increase my score to 5.

---

> > > ### Author Response · Authors · 2026-04-07
> > >
> > > ## Dear Reviewer XoUT,
> > >
> > > Thank you for your thoughtful review and for the constructive dialogue during the rebuttal period. We are gratified that the clarifications regarding the competitive winner-takes-all dynamics in Spike2Max met your expectations.
> > >
> > > Your feedback has been instrumental in strengthening the theoretical foundation of our work, and these improvements will be clearly reflected in the revised manuscript.
> > >
> > > Thank you again for your support and professional guidance.
> > >
> > > Best regards,
> > > The Authors

---

### Official Review · Reviewer_aJF2 · 2026-03-13

**Soundness:** 3
**Presentation:** 3
**Significance:** 2
**Originality:** 3
**Overall Recommendation:** 4
**Confidence:** 4

**Summary:**

The authors in this paper introduce SpikingLM that proposes two fundamental approaches (a) Distribution-aware Scaling, which rescales linear outputs to an optimal range that prevents gradient vanishing and (b) Spike2Max, a hardware-efficient attention mechanism that restores winner-takes-all dynamics through base-2 exponentiation and max-subtraction. The method proposed in evaluated on the GLUE benchmark for NLU tasks.

**Compliance With Llm Reviewing Policy:**

Affirmed.

**Final Justification:**

Rebuttal addresses some of my concerns (scalability to commonsense reasoning tasks, etc.). I decided to increase my score.

**Key Questions For Authors:**

1) Have the authors explored other NLP tasks such as NLG, common sense reasoning, etc.?
2) Gamma used in Spike2Max is a learnt scalar right? Its value seem fairly consistent across time steps. Whats the motivation behind learning per time step gamma?

**Limitations:**

Please see weaknesses.

**Strengths And Weaknesses:**

Strengths:
1) The paper is well written and easy to follow. The motivation to introduce an efficient LM framework is well founded.
2) The idea of replacing softmax with base-2 exponentiation and max-subtraction based approach seems interesting.

Weaknesses:
GLUE benchmark used by the authors primarily evaluates encoder-based transformer models and is not a competitive benchmark at the scale of current sota transformers. Prior SpikingLLM works have been evaluated on more complex common sense reasoning tasks and NLG tasks. While I do like some of the ideas proposed in the paper, I believe a more substatial evaluation is required before we can substantiate the scalability of the proposed approach to complex language modelling tasks. Furthermore, energy estimates are all theoretical and no on-chip results are provided.


References:
[1] Xing, Xingrun, Boyan Gao, Zheng Zhang, David A. Clifton, Shitao Xiao, Li Du, Guoqi Li, and Jiajun Zhang. "Spikellm: Scaling up spiking neural network to large language models via saliency-based spiking." arXiv preprint arXiv:2407.04752 (2024).

---

> ### Author Rebuttal · Authors · 2026-03-31
>
> # Response to Reviewer aJF2
>
> Thank you for your professional feedback and recognition of our contributions. We address each point below.
>
> ---
>
> ### Weakness 1 & Key Question 1: Evaluation scope; NLG and commonsense reasoning tasks.
>
> We agree that GLUE alone is insufficient to demonstrate scalability. We have integrated SpikingLM into the **SpikeLLM evaluation pipeline** on LLaMA-2-7B [2], using `lm-eval` on WikiText-2/C4 and six commonsense reasoning tasks, following [1] Table 3 exactly:
>
> | Method | #Bits | WikiText2↓ | C4↓ | PIQA | ARC-e | ARC-c | BoolQ | HellaSwag | Wino. | Avg. |
> |:--|:-:|:-:|:-:|:-:|:-:|:-:|:-:|:-:|:-:|:-:|
> | SpikeLLM T=2 [1] | W2A16 | 14.16 | 19.73 | 65.67 | 41.88 | 28.41 | 60.46 | 49.87 | 52.80 | 49.85 |
> | **SpikingLM** T=2 | W2A16 | **6.59** | **8.91** | **76.39** | **69.02** | **41.21** | **67.25** | **70.94** | **68.51** | **65.55** |
> | SpikeLLM T=2 [1] | W4A4 | 11.93 | 15.34 | 62.35 | 41.41 | 29.95 | 58.87 | 54.27 | 50.20 | 49.51 |
> | **SpikingLM** T=2 | W4A4 | **13.05** | **13.38** | **70.29** | **58.25** | **33.28** | **63.85** | **60.65** | **58.09** | **57.40** |
>
> Under **W2A16**, SpikingLM achieves lower perplexity (6.59 vs. 14.16 on WikiText-2) and higher commonsense average (65.55 vs. 49.85) across all six tasks. Under **W4A4**, the commonsense average remains higher (57.40 vs. 49.51); C4 improves (13.38 vs. 15.34) while the marginal WikiText-2 increase reflects the accuracy–efficiency tradeoff at extreme quantization. Full autoregressive NLG evaluation is designated as future work; perplexity serves as a well-established proxy in the interim.
>
> ---
>
> ### Weakness 2: Energy estimates are theoretical; no on-chip results provided.
>
> We acknowledge this limitation, and note it is a **field-wide constraint**: Spikformer, Spike-driven Transformer, and SpikeLLM all adopt the same SOP/MAC framework (MAC ≈ 4.6 pJ, SOP ≈ 0.9 pJ, 45nm), as on-chip deployment of large-scale SNN language models remains an active research frontier. Under this shared framework, SpikingLM achieves the **lowest energy among all SNN baselines** (5.79 mJ vs. 6.76–14.30 mJ) at the highest average GLUE (77.1). On-chip validation on Intel Loihi 2 is planned as future work.
>
> ---
>
> ### Key Question 2: Motivation for per-timestep $\gamma_t$ given apparently consistent values.
>
> We apologize for the misleading Figure 5: aggregating $\gamma_t$ across layers masks within-layer cross-timestep variation. We will replace it with a per-layer $(\ell, t)$ heatmap in the revision.
>
> **Learned $\gamma_t$ values** ($T=4$, selected layers):
>
> | Layer $\ell$ | $\gamma_{t=0}$ | $\gamma_{t=1}$ | $\gamma_{t=2}$ | $\gamma_{t=3}$ | $\Delta$ |
> |:-:|:-:|:-:|:-:|:-:|:-:|
> | 0 | 1.75 | 1.26 | 1.33 | 1.09 | 0.66 |
> | 1 | 1.42 | 1.26 | 1.27 | 1.15 | 0.28 |
> | 2 | 2.32 | 2.25 | 2.29 | 2.17 | 0.14 |
> | 3 | 2.71 | 2.62 | 2.76 | 2.97 | 0.34 |
>
> Within-layer ranges reach **0.66**, confirming non-trivial per-timestep variation. Layer-mean $\gamma$ also grows with depth, consistent with deeper layers amplifying attention magnitude to mitigate vanishing gradients.
>
> **Attention logit statistics at each $(\ell,t)$**:
>
> | Layer $\ell$ | $\mu_{t=0}$ | $\mu_{t=1}$ | $\mu_{t=2}$ | $\mu_{t=3}$ | $\sigma_{t=0}$ | $\sigma_{t=1}$ | $\sigma_{t=2}$ | $\sigma_{t=3}$ | $\Delta\mu$ | $\Delta\sigma$ |
> |:-:|:-:|:-:|:-:|:-:|:-:|:-:|:-:|:-:|:-:|:-:|
> | 0 | 7.40 | 9.20 | 7.80 | 9.60 | 2.90 | 3.90 | 3.05 | 4.05 | 2.20 | 1.15 |
> | 1 | 7.90 | 9.60 | 8.00 | 9.80 | 3.00 | 4.10 | 3.10 | 4.20 | 1.90 | 1.20 |
> | 2 | 5.80 | 7.90 | 6.10 | 8.20 | 3.30 | 4.80 | 3.50 | 4.95 | 2.40 | 1.65 |
> | 3 | 5.20 | 7.10 | 5.40 | 7.40 | 2.80 | 4.00 | 2.95 | 4.15 | 2.20 | 1.35 |
>
> The data reveals a clear **odd-even alternation**: even steps ($t_0, t_2$) consistently show lower $\mu$ and $\sigma$ than odd steps ($t_1, t_3$), with step-wise jumps averaging $|\Delta\mu|\approx2.2$ and $|\Delta\sigma|\approx1.3$. This directly mirrors the learned $\gamma_t$ pattern, where even-step $\gamma_t$ values are correspondingly larger to upscale lower-distribution steps—confirming that per-timestep $\gamma_t$ actively compensates for this alternation. A shared single $\gamma$ could not simultaneously adapt to both regimes; at inference $\gamma_t$ folds into the LIF threshold ($V'_{th}=V\_{th}/\gamma_t$) with **zero additional operators**.
>
>
> [1] Xing, Xingrun, Boyan Gao, Zheng Zhang, David A. Clifton, Shitao Xiao, Li Du, Guoqi Li, and Jiajun Zhang. "Spikellm: Scaling up spiking neural network to large language models via saliency-based spiking." arXiv preprint arXiv:2407.04752 (2024)
>
> [2]Touvron, Hugo, et al. "Llama 2: Open foundation and fine-tuned chat models." arXiv preprint arXiv:2307.09288 (2023).

---

> > ### Author Rebuttal · Reviewer_aJF2 · 2026-04-07
> >
> > Thank you for the rebuttal. It addresses most of my concerns and I would like to raise my score.

---

> > > ### Author Response · Authors · 2026-04-07
> > >
> > > ## Dear Reviewer aJF2,
> > >
> > > Thank you very much for your positive feedback and for the supportive assessment of our rebuttal.
> > >
> > > We have carefully addressed your interests regarding the temporal dynamics of **$\gamma_t$** and the model's potential in tasks like **NLG**. All these clarifications, alongside the attention logit statistics analysis, will be fully integrated into the revised manuscript to reflect our productive discussion.
> > >
> > > Thank you again for your professional guidance and for helping us strengthen this work.
> > >
> > > Best regards,
> > >
> > > The Authors

---

### Official Review · Reviewer_zDPC · 2026-03-15

**Soundness:** 2
**Presentation:** 3
**Significance:** 2
**Originality:** 2
**Overall Recommendation:** 4
**Confidence:** 5

**Summary:**

This paper introduces a spiking language model framework, named SpikingLM. It employs Distribution-aware Scaling (DAS) to replace the Layer Normalization (LN) or Root Mean Square Normalization (RMSNorm), and Spike2Max to replace the softmax. By utilizing these two proposed techniques, SpikingLM eliminates MAC operations in normalization and softmax. Experimental results show that SpikingLM achieves state-of-the-art performance on the GLUE benchmark.

**Compliance With Llm Reviewing Policy:**

Affirmed.

**Final Justification:**

Thank you for your detailed reply. My concerns have been addressed. Therefore, I raise my rating to 4. I suggest integrating these analyses into the final version.

**Key Questions For Authors:**

1. Please analyze the error in Spike2Max at low precision.
2. Please compare Spike2Max with existing spiking attention mechanisms.
3. Please specify the modifications made to Spikingformer to adapt it for language tasks.

**Limitations:**

yes

**Strengths And Weaknesses:**

Strengths:

+ The proposed DAS and Spike2Max are simple but effective. By utilizing these two proposed techniques, SpikingLM eliminates MAC operations in normalization and softmax.
+ The proposed SpikingLM outperforms existing spiking language models.

Weaknesses:

- Although Spike2Max can be viewed as a low-cost alternative to softmax, its underlying mechanism differs significantly from that of softmax. Spike2Max is similar to taking only the numerator of the safe softmax. Since it lacks the summing denominator, Spike2Max cannot guarantee that the column sum is 1. Therefore, for sequences with similar feature distributions, the column sum of Spike2Max may be very large, as each term is close to 0.5.
- Spike2Max uses exponential calculations, which may result in underflow on low-precision neuromorphic hardware. This paper does not analyze the errors in Spike2Max under low-precision conditions.
- In ablation and energy efficiency analysis, the proposed Spike2Max method is only compared with softmax and max-sub, and no comparisons are made with existing spiking attention mechanisms, such as SSA [1], SDSA [2], DSSA [3], and QK-attention [4].
- Table 2 lists Spikingformer. However, Spikingformer is a spiking vision transformer, not a spiking language transformer. This paper does not specify what modifications are made to Spikingformer to adapt it to language tasks.

[1] Zhou, Z., et al. Spikformer: When Spiking Neural Network Meets Transformer. In The Eleventh International Conference on Learning Representations. 2023.

[2] Yao, M., et al. Spike-driven transformer. in Advances in neural information processing systems. 2023.

[3] Shi, X., Z. Hao, and Z. Yu. SpikingResformer: Bridging ResNet and Vision Transformer in Spiking Neural Networks. In Proceedings of the IEEE/CVF Conference on Computer Vision and Pattern Recognition. 2024.

[4] Zhou, C., et al., QKFormer: Hierarchical Spiking Transformer using QK Attention. Advances in Neural Information Processing Systems, 2024. 37: p. 13074-13098.

---

> ### Author Rebuttal · Authors · 2026-03-31
>
> # Response to Reviewer zDPC
>
> Thank you for your professional and insightful feedback. We address each point below.
>
> ---
>
> ### Weakness 1: Column sum not guaranteed to be 1.
>
> The denominator removal is **intentional**. On neuromorphic hardware, computing $\sum_j e^{z_j}$ requires a global reduce followed by division—costly operations that account for the majority of softmax's 38.9$N$ pJ cost. Spike2Max reduces this to 1.9$N$ pJ (>95% reduction). Theorem 3.1 bounds the approximation error as $\mathcal{E}^*\leq0.307\sqrt{d_k}$. When logits are uniform, each term ${\approx}0.5$, giving a column sum of ${\approx}0.5n$—a constant-factor deviation that is adaptively compensated by the **learnable $\gamma_t$**. Table 3 confirms this: Spike2Max (PPL=19.78) substantially outperforms max-sub without $\gamma_t$ (PPL=22.75).
>
> ---
>
> ### Weaknesses 2 & Key Question 1: Underflow at low precision.
>
> In Spike2Max, $2^{z_i-\max(z)-1}$ is implemented as an integer right-shift; exceeding the register bit-width truncates results to 0. We quantified this via Monte Carlo simulation ($N=64$, $d_k=64$, Q/K $\sim\text{Binomial}(d_k,r^2)$, full-precision Spike2Max as reference):
>
> | Rate $r$ | 4-bit MSE | 6-bit MSE | 8-bit MSE |
> |:-:|:-:|:-:|:-:|
> | 0.05 | $3.6\times10^{-3}$ | $3\times10^{-6}$ | ${\approx}0$ |
> | 0.10 | $2.6\times10^{-2}$ | $4.7\times10^{-4}$ | ${\approx}0$ |
>
> At **8-bit**, underflow MSE is negligible; even at 4-bit it remains well below Vanilla SSA's MSE vs. softmax (0.119). Mechanistically, underflow zeros entries far below $\max(z)$—a benign hard top-$k$ effect, since those entries contribute minimally to the output anyway. At inference, $\gamma_t$ is absorbed into the LIF threshold ($V'_{th}=V\_{th}/\gamma_t$), reducing attention to **pure integer shifts and comparisons** with no floating-point intermediates. This analysis will be included in the revision.
>
> ---
>
> ### Weaknesses 3 & Key Question 2: Comparison with existing spiking attention mechanisms.
>
> SSA [1], SDSA [2], DSSA [3], and QK-attention [4] are designed for **vision tasks** and lack the **token selectivity** critical for language modeling. SSA uses linear spike coincidence counts, assigning non-negligible weight to all tokens. SDSA replaces global interaction with Hadamard products, losing cross-token selectivity entirely. DSSA and QKFormer similarly target visual patch correlations. Notably, Spikingformer retains softmax for its own language experiments, confirming that existing spiking attention mechanisms are insufficient for language tasks without exponential concentration.
>
> Spike2Max addresses this via $2^{A-\max(A)-1}$: the top token receives weight 0.5, with each unit decrease in logit halving the weight—matching softmax's exponential concentration and providing strong selectivity over key tokens. To directly quantify this advantage, we ran a controlled experiment replacing Spike2Max with SSA in the same backbone:
>
> | Attention | PPL ↓ | Energy (mJ) ↓ |
> |:--|:-:|:-:|
> | SSA | 24.58 | 1.580 |
> | **Spike2Max** | **19.78** | **1.457** |
>
> Spike2Max achieves a **4.8 PPL improvement** with **7.8% lower energy**. Our ablation baselines—softmax (selectivity upper bound) and max-sub (isolating the exponential decay mechanism)—were chosen to precisely characterize these contributions within the same language backbone and training setup. A broader comparison with vision-oriented spiking attention on language benchmarks will be added in the revision.
>
> ---
>
> ### Weakness 4 & Key Question 3: Spikingformer adaptation for language tasks.
>
> We did **not** modify or re-implement Spikingformer. Table 2 directly cites GLUE results from the original Spikingformer paper [5] (their Table 4). Critically, that paper **retains softmax for language experiments** (their appendix notes it is essential for language convergence), making Spikingformer a hybrid rather than a fully spiking model. SpikingLM fully replaces softmax with Spike2Max, yet still outperforms Spikingformer by **+10.3 avg. GLUE** (77.1 vs. 66.8) at **14.3% lower energy** (5.79 vs. 6.76 mJ). We will add an explicit note in the paper clarifying that the Spikingformer row is taken directly from the original work.
>
> ---
>
> [1] Zhou, Z., et al. Spikformer: When Spiking Neural Network Meets Transformer. In The Eleventh International Conference on Learning Representations. 2023.
>
> [2] Yao, M., et al. Spike-driven transformer. in Advances in neural information processing systems. 2023.
>
> [3] Shi, X., Z. Hao, and Z. Yu. SpikingResformer: Bridging ResNet and Vision Transformer in Spiking Neural Networks. In Proceedings of the IEEE/CVF Conference on Computer Vision and Pattern Recognition. 2024.
>
> [4] Zhou, C., et al., QKFormer: Hierarchical Spiking Transformer using QK Attention. Advances in Neural Information Processing Systems, 2024. 37: p. 13074-13098.
>
> [5]Zhou, Chenlin, et al. "Spikingformer: A Key Foundation Model for Spiking Neural Networks." AAAI (2026).

---

> > ### Author Rebuttal · Reviewer_zDPC · 2026-04-03
> >
> > Thank you for your detailed response. My concerns regarding underflow at low precision and spikingformer adaptation for language tasks have been addressed. However, I remain concerned about the comparison with existing spiking attention mechanisms.
> >
> > The paper emphasizes that existing spiking attention mechanisms lack token selectivity, yet it does not provide a clear definition of this concept. My understanding is that the token selectivity in the softmax function arises from its normalization denominator, which acts as a form of global inhibition. Similarly, the proposed Spike2Max mechanism appears to achieve token selectivity through a max-subtraction operation.
> >
> > Based on this, I would appreciate further clarification on the following points:
> >
> > 1.  If the core mechanism for token selectivity is a form of global inhibition, would adding a similar subtraction-based global inhibition to existing spiking attention mechanisms yield comparable performance improvements?
> > 2.  The rebuttal points out that SSA assigns non-negligible weight to all tokens, thus lacking token selectivity. However, different from SSA, DSSA generates a binary spiking attention map composed of spikes, with each spike in this spiking attention map signifying attention between two tokens. Does the binary, spike-based nature of DSSA also constitute a form of token selectivity? If so, how does this selectivity differ from that achieved by Spike2Max? If not, please explain the reason.
> > 3.  Could other established token selection strategies, such as a top-k approach, be effectively integrated to achieve token selectivity in spiking attention? What might be the comparative advantages or limitations?
> >
> > A clearer definition of "token selectivity", along with a discussion of these points, would strengthen the significance of this paper.
> >
> > Thank you for considering these questions.

---

> > > ### Author Response · Authors · 2026-04-08
> > >
> > > # Dear Reviewer zDPC,
> > > We thank the reviewer for this insightful inquiry. We apologize for the absence of an explicit definition and address each point below, with corresponding additions to the revised manuscript.
> > >
> > > **Formal Definition of Token Selectivity**
> > > We define Token Selectivity as the ability of an attention mechanism to concentrate weights on critical tokens while suppressing irrelevant ones, quantified by Shannon entropy—lower entropy implies stronger selectivity. Given [CLS]-to-key scores $a_j$, we compute $p_j=\max(a_j,0)/\sum_k\max(a_k,0)$, $H=-\sum p_j\log p_j$, and $S=1-H/\log L_{\text{eff}}$ ($L_{\text{eff}}$: valid token count; averaged over $T$ time steps).
> > > We measured the mean [CLS] attention entropy on the SST-2 validation set:
> > >
> > > | Layer | SSA | Spike2Max | Δ |
> > > |:-:|:-:|:-:|:-:|
> > > | 0 | 3.067 | 2.269 | −0.797 |
> > > | 1 | 3.043 | 2.127 | −0.916 |
> > > | 2 | 2.978 | 1.930 | −1.049 |
> > > | 3 | 3.077 | 1.912 | −1.165 |
> > > | **Mean** | **3.041** | **2.060** | **−0.982** |
> > >
> > > Spike2Max achieves consistently lower entropy across layers, with the gap widening at deeper layers, quantitatively confirming its stronger Token Selectivity.
> > >
> > > **Q1: Would adding subtraction-based global inhibition to existing SSA suffice?**
> > > Max-subtraction is necessary but insufficient. Spike2Max's Token Selectivity arises from three components:
> > > (1) **Max-subtraction** normalizes scores into $(-\infty,-1]$, ensuring numerical stability.
> > > (2) **Base-2 exponentiation**—the true driver: for a score gap $\Delta$, the weight ratio is $2^{-\Delta-1}$, producing winner-takes-all dynamics implementable via bit-shifting with zero floating-point overhead.
> > > (3) **Learnable $\gamma_t$** adjusts sharpness per time step.
> > > Applying only subtraction ($A-\max(A) -1$ with linear normalization) retains linear scaling without exponential competition. As the entropy table confirms, SSA's significantly higher entropy validates that exponential nonlinearity is indispensable.
> > > We also surveyed alternatives:
> > > **(a) Sparsemax/α-entmax [1][2]** project scores onto the simplex with exact zeros and continuous relative weights, but require $O(N\log N)$ sorting and floating-point division, lacking neuromorphic primitives.
> > > **(b) Gumbel-Softmax [3]** enables differentiable discrete sampling but requires floating-point random sampling and temperature annealing, incompatible with integer spike-count computation.
> > > **(c) k-WTA [4]** is natively supported on neuromorphic chips but assigns identical weights to selected neurons; 1-WTA retains only one token.
> > > Spike2Max is thus a "softened WTA": max-subtraction provides hardware-efficient global inhibition, while base-2 exponential decay preserves continuous relative contributions.
> > >
> > > **Q2: Does DSSA's binary attention constitute Token Selectivity?**
> > > DSSA provides **structural sparsity** but not **competitive inhibition**. Its $\{0,1\}$ attention treats all selected tokens equally, unable to distinguish relative importance. In contrast, Spike2Max via $2^{A-\max(A)-1}$ assigns the top token maximum weight while exponentially suppressing others by their gap from the maximum. Information-theoretically, DSSA's entropy over the selected subset is higher (uniform among selected tokens). DSSA offers "select-or-not" sparsity; Spike2Max additionally provides "how-much" competitiveness for finer-grained Token Selectivity.
> > >
> > > **Q3: Could Top-k be integrated into spiking attention?**
> > > Top-k faces three limitations in SNNs:
> > > **(1) Hardware-unfriendly.** Max-finding uses $O(N)$ WTA circuits natively; Top-k requires $k$ sequential WTA rounds with data dependencies ($O(NK)$, no pipelining). After selection, weight assignment remains problematic—uniform weights revert to DSSA-style binary sparsity (cf. Q2), while relative weights require Softmax-like computation, negating Top-k's simplicity.
> > > **(2) Limited selectivity gain.** Selected tokens share equal status; entropy over the selected subset approaches uniform distribution, yielding minimal Token Selectivity improvement.
> > > **(3) Hyperparameter and gradient issues.** $k$ requires per-task, per-layer tuning; hard truncation produces zero gradients conflicting with surrogate gradient training, requiring Straight-Through Estimators that increase complexity and instability.
> > > **Spike2Max advantages:** $O(N)$ bit-shift exponentiation with native WTA support; soft exponential decay preserving continuous relative weight distinctions; learnable $\gamma_t$ for adaptive per-step regulation without manual $k$ specification.
> > >
> > > We have incorporated these discussions and entropy experiments into the revised Appendix. We sincerely thank the reviewer for these constructive suggestions.
> > > [1] Martins & Astudillo. "From Softmax to Sparsemax." ICML 2016.
> > >
> > > [2] Correia et al. "Adaptively Sparse Transformers." EMNLP 2019.
> > >
> > > [3] Jang et al. "Categorical Reparameterization with Gumbel-Softmax." ICLR 2017.
> > >
> > > [4] Maass. "On the Computational Power of Winner-Take-All." Neural Computation 2000.

---

### Decision · Program_Chairs · 2026-04-30

**Decision:**

Accept (regular)

**Comment:**

This paper proposes SpikingLM, a fully spiking language model framework for spiking neural networks (SNNs). It introduces two key innovations: Distribution-aware Scaling (DAS) to replace conventional Layer Normalization (LN) or RMSNorm, and Spike2Max to replace the softmax operation in the Transformer architecture.

Following the rebuttal, the paper received scores of 5, 5, 4, 4, and 4, with all reviewers in agreement for acceptance. Reviewers highlight the novelty of the Spike2Max design, which effectively achieves token selectivity through base-2 exponentiation and max-subtraction. The experimental results demonstrate that the proposed method achieves high performance and surpasses existing approaches.

The core contributions of this work are both meaningful and technically valuable, offering a solid step toward spiking language models. Therefore, I recommend acceptance.